# Closing the Train-Test Gap in World Models for Gradient-Based Planning

## Abstract

World models paired with model predictive control (MPC) can be trained offline on large-scale datasets of expert trajectories and enable generalization to a wide range of tasks chosen at inference time. Compared to traditional MPC procedures, which rely either on slow search algorithms or on iteratively solving optimization problems exactly, gradient-based planning offers a computationally efficient alternative. However, the performance of gradient-based planning has thus far lagged behind that of other approaches. In this paper, we propose improved methods for training world models that enable efficient gradient-based planning. We begin with the observation that although a world model is trained on a next-state prediction objective, it is used at test-time to instead estimate a sequence of actions. The goal of our work is to close this train-test gap. To that end, we propose train-time data synthesis techniques that enable significantly improved gradient-based planning with existing world models. At test time, our approach outperforms or matches the classical gradient-free cross-entropy method (CEM) across a variety of object manipulation and navigation tasks in 10% of the time budget.

## 1 Introduction

In robotic tasks, anticipating how the actions of an agent affect the state of its environment is fundamental for both prediction (Finn et al., 2016) and planning (Mohanan & Salgoankar, 2018; Kavraki et al., 2002). Classical approaches derive models of the environment evolution analytically from first principles, relying on prior knowledge of the environment, the agent, and any uncertainty (Goldstein et al., 1950; Siciliano et al., 2009; Spong et al., 2020). In contrast, learning-based methods extract such models directly from data, enabling them to capture complex dynamics and thus improve generalization and robustness to uncertainty (Sutton et al., 1998; Schrittwieser et al., 2020; LeCun, 2022).

World models (Ha & Schmidhuber, 2018), in particular, have emerged as a powerful paradigm. Given the current state and an action, the world model predicts the resulting next state. These models can be learned either from exact state information (e.g. Sutton, 1991) or directly from high-dimensional sensory inputs such as images (e.g. Hafner et al., 2023) . The latter is especially compelling as it enables perception, prediction, and control directly from raw images by leveraging pretrained visual representations, and removes the need for measuring the precise environment states which is difficult in practice (Assran et al., 2023; Bardes et al., 2024).

Recently, world models have been shown to leverage their predictive capabilities for planning, enabling agents to solve a variety of tasks (Hafner et al., 2019a;b; Schrittwieser et al., 2020; Hafner et al., 2023; Zhou et al., 2025). A model of the dynamics is learned offline, while the planning task is defined at inference as a constrained optimization problem: given the current state, find a sequence of actions that comes as close as possible to a target state. Because the planning objective can be specified at test time, training the world model does not need to be task-specific; the same model can be reused across different tasks simply by modifying the planning objective. This inference-time optimization provides an effective alternative to reinforcement learning (RL) approaches (Sutton et al., 1998). Unlike model-free RL, which often suffers from poor sample-efficiency, or model-based RL (Hansen et al., 2023; Hafner et al., 2023), which typically requires training a separate policy for each new task, planning with world models can evaluate potential actions without interacting with

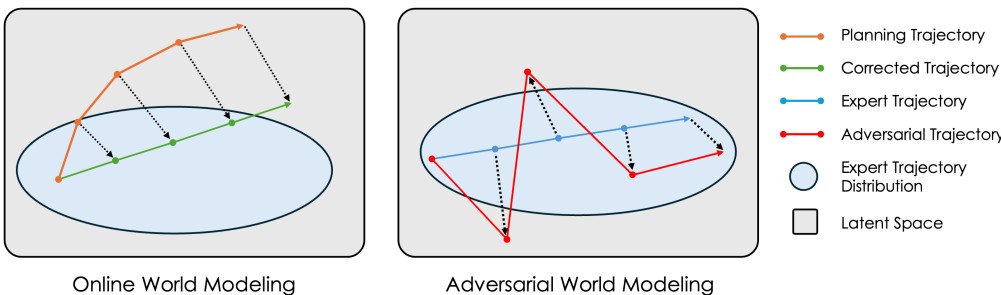

Figure 1: An overview of our two proposed methods. Online World Modeling finetunes a pretrained world model using the simulator dynamics function $h$ to correct trajectories produced during planning that may exit the expert trajectory distribution. Adversarial World Modeling pretrained finetunes a world model on perturbations of expert trajectories such that they may exit the expert trajectory distribution and promote robustness in the world model during planning.

the environment and specifies the task only at inference time, enabling efficient generalization across multiple tasks.

Several model-based planning algorithms can be used with world models. Traditional methods, such as DDP (Mayne, 1966) and iLQR (Li & Todorov, 2004), rely on iteratively solving exact optimization problems derived from linear and quadratic approximations of the dynamics around a nominal trajectory. While highly effective in low-dimensional settings, these methods become impractical for large-scale world models, where solving the resulting optimization problem is computationally intractable. As an alternative, search-based methods such as the Cross Entropy Method (CEM) (Rubinstein & Kroese, 2004) and Model Predictive Path Integral control (MPPI) (Williams et al., 2017a) have been widely adopted as a gradient-free alternative and have proven effective in practice. However, they are computationally intensive as they require iteratively sampling candidate solutions and performing world model rollouts to evaluate each one, a procedure that scales poorly in high-dimensional spaces. Gradient-based methods (SV et al., 2023), by contrast, avoid the limitations of sampling by directly exploiting the differentiability of world models. These methods eliminate the costly rollouts required by search-based approaches, thus scaling more efficiently in high-dimensional spaces. However, they have seen little success to date.

World models have a fundamental train-test gap. Their training objective is next-state prediction along expert trajectories, whereas at test time, planning involves solving an optimization problem over multiple consecutive actions. We offer two hypotheses for why this train-test gap causes poor performance in gradient-based planning (GBP): **(1)** The sub-optimal sequence of actions being optimized over force the world model into states it has not encountered during planning. Poor performance of the world model on these states makes it unreliable for optimizing through. **(2)** The actions loss surface is difficult to optimize over and contains many local minima, consequently hindering gradient descent during GBP.

In this work, we address these challenges by proposing two algorithms: Online World Modeling and Adversarial World Modeling. Both expand the region of familiar latent states by continuously adding new trajectories to the dataset and finetuning the world model on them, shown in Figure 1. Online World Modeling is based on imitation learning and attempts to correct the distribution shift between expert and planning trajectories. Adversarial World Modeling trains the world model on adversarial perturbations of trajectories, smoothing the planning loss surface in the process.

We demonstrate that using these algorithms to finetune a world model results in significantly improved GBP performance. Notably, Adversarial World Modeling with GBP matches or outperforms the search-based CEM with a pretrained world model on object manipulation and navigation tasks, while being significantly more computationally efficient. Furthermore, we present evidence of our algorithms making progress towards alleviating the train-test gap.

## 2 RELATED WORK

**Learning world models from sensory data.** Learning-based dynamics models have become central to control and decision making, offering a data-driven alternative to classical approaches that rely on first principles modeling (Goldstein et al., 1950; Schmidt & Lipson, 2009; Macchelli et al., 2009). Early work focused on modeling dynamics in low-dimensional state-space (Deisenroth & Rasmussen, 2011; Lenz et al., 2015; Henaff et al., 2017; Sharma et al., 2019), while more recent methods learn directly from high-dimensional sensory inputs such as images. One line of work trains models to predict future observations in pixel-space (Finn et al., 2016; Kaiser et al., 2019), demonstrating success in applications such as human motion prediction (Finn et al., 2016), robotic manipulation (Finn & Levine, 2016; Agrawal et al., 2016; Zhang et al., 2019), and solving Atari games (Kaiser et al., 2019). However, pixel-level prediction is often computationally expensive due to the cost of reconstructing images. To address this, alternative approaches learn a compact latent representation where dynamics are modeled (Karl et al., 2016; Hafner et al., 2019b; Shi et al., 2022; Karypidis et al., 2024). These models are typically supervised either by decoding latent predictions to match ground truth observations (Edwards et al., 2018; Zhang et al., 2021; Bounou et al., 2021; Hu et al., 2022; Akan & Güney, 2022; Hafner et al., 2019b), or by using prediction objectives that operate directly in latent space, such as those in joint-embedding prediction architectures (JEPAs) (LeCun, 2022; Bardes et al., 2024; Drozdov et al., 2024; Guan et al., 2024; Zhou et al., 2025). Our method builds upon this latter category of world models, and specifically leverages the approach introduced by Zhou et al. (2025).

**Planning with world models.** Planning with world models is challenging due to their inherent non-linearity and non-convexity. Search-based methods such as CEM (Rubinstein & Kroese, 2004) and MPPI (Williams et al., 2017a) are widely used in this context (Williams et al., 2017b; Nagabandi et al., 2019; Hafner et al., 2019b; Zhan et al., 2021; Zhou et al., 2025). These methods explore the action space effectively, helping to escape from local minima, but typically scale poorly in high dimensions because of their sampling nature. In contrast, gradient-based methods exploit the differentiability of the world model to optimize actions directly via backpropagation. This approach offers better scalability, but suffers from local minima, adversarial trajectories, and non-smooth objective landscapes (Bharadhwaj et al., 2020a; Xu et al., 2022; Chen et al., 2022; Wang et al., 2023). To combine the strengths of both approaches, hybrid methods have been proposed. For example, (Bharadhwaj et al., 2020a) interleave CEM and gradient descent steps during optimization, leveraging CEM for global exploration and gradient descent for local refinement. In this work, we focus on improving the planning capabilities of world models. Zhou et al. (2025) show that when using DINOv2 (Oquab et al., 2024) embeddings, gradient-based planning underperforms compared to CEM. We build on this approach, focusing on improving gradient-based planning performance.

**Train-test gap in planning with world models** A key challenge when planning with learned world models is the mismatch between training distributions and the trajectories generated during test-time planning (Ajay et al., 2018; Ke et al., 2019; Zhu et al., 2023). In face, models are typically trained to minimize prediction or reconstruction error on trajectories from a dataset or a behavioral policy, but planning algorithms can generate at test-time out-of-distribution action sequences that drive the model into poorly-trained regions, potentially leading to compounding errors or adversarial trajectories that exploit model inaccuracies (Schiewer et al., 2024; Jackson et al., 2024). Several strategies aim to address this train-test gap. Techniques like random-shooting can further help mitigate adversarial trajectories (Nagabandi et al., 2018). Alternatively, regularization-based methods, such as implicit policy training with gradient penalties, aim to improve model smoothness and planning stability (Florence et al., 2022). Closer to our approach, dataset-aggregation methods (Ross et al., 2011) expand the training distribution by rolling out action trajectories found by the planning algorithm and adding them to the training set (Talvitie, 2014; Nagabandi et al., 2018). In a similar spirit to our approach, Zhang et al. (2025) introduce an adversarial attack method to encourage diverse state visitation distribution in a model-based RL setting.

## 3 WORLD MODELS AND GRADIENT-BASED PLANNING

We present two data aggregation methods for closing the train-test gap between the standard world modeling objective and gradient-based planning: Online World Modeling and Adversarial World Modeling. We provide a visual depiction of both methods in Figure 1.

## 3.1 PROBLEM FORMULATION

World models learn environment dynamics via next-state prediction on raw observations. At test time, the learned model enables planning by simulating future trajectories and serving as a dynamics constraint during action optimization.

Let $\mathcal{S}$ denote the state space and $\mathcal{A}$ the action space. The environment evolves following a typically unknown dynamics function $h$ such that

$$h : \mathcal{S} \times \mathcal{A} \to \mathcal{S}, \quad s_{t+1} = h(s_t, a_t), \quad \text{for all } t, \tag{1}$$

where $s_t, a_t$ denote the state and action at time $t$, respectively. In practice, we only have access to a sequence $[o_1, \ldots, o_T]$, where $o_t \in \mathcal{O} \subset \mathbb{R}^p$ is an observation of $s_t$. Following Zhou et al. (2025), we learn a dynamics model on a compact embedding of the world. Given an embedding function $\Phi : \mathcal{O} \to \mathcal{Z}$, our goal is to learn a latent world model $f_\theta : \mathcal{Z} \times \mathcal{A} \to \mathcal{Z}$, such that

$$z_t = \Phi(o_t), \quad z_{t+1} = f_\theta(z_t, a_t), \quad \text{for all } t. \tag{2}$$

**Encoder.** Following Zhou et al. (2025), we use a pretrained encoder as our embedding function $\Phi$. Pretraining on a wide range of visual domains yields rich feature representations, enabling the latent world model to generalize robustly across different tasks.

**World model.** Given an environment, we train a world model with the teacher-forcing objective

$$\min_\theta \mathbb{E}_{(o_t, a_t, o_{t+1})} \| f_\theta(\Phi_\mu(o_t), a_t) - \Phi_\mu(o_{t+1}) \|_2^2. \tag{3}$$

The triplets $(o_t, a_t, o_{t+1})$ are sampled from an offline dataset of trajectories and we minimize the $\ell_2$ distance between the true and predicted embeddings of the next state $o_{t+1}$.

**Planning.** The planning objective is defined at test-time. Given an initial state $z_1 \in \mathcal{Z}$ and a goal state $z_{\text{goal}} \in \mathcal{Z}$, the planning task is to find a sequence of actions $\{\hat{a}_t^*\}_{t=1}^H$ that drives the system to the goal. Formally, we solve

$$\{\hat{a}_t^*\}_{t=1}^H = \arg\min_{\{\hat{a}_t\}} \| \hat{z}_{H+1} - z_{\text{goal}} \|_2^2 \tag{4}$$

where the latent trajectory is generated recursively as

$$\hat{z}_2 = f_\theta(z_1, \hat{a}_1), \quad \hat{z}_{t+1} = f_\theta(\hat{z}_t, \hat{a}_t) \quad \text{for} \quad t > 1. \tag{5}$$

We denote this recursive procedure with the function $\text{rollout}_f : \mathcal{Z} \times \mathcal{A}^H \to \mathcal{Z}^H$. Gradient-based planning (GBP) solves the planning task via minimizing the loss function $\| \hat{z}_{H+1} - z_{\text{goal}} \|_2^2$ with respect to the sequence of actions $\{\hat{a}_t\}$ via gradient descent. Crucially, since the world model is differentiable, $\nabla_{\{\hat{a}_t\}} \hat{z}_{H+1} = \nabla_{\{\hat{a}_t\}} \text{rollout}_f(z_1, \{\hat{a}_t\})_{H+1}$ is well-defined. We describe GBP in detail in Algorithm 1. As errors can propagate over long horizons, Model Predictive Control (MPC) is commonly used to repeatedly re-plan by optimizing an $H$-step action sequence but executing only the first $K \leq H$ actions before replanning from the updated state.

---

**Algorithm 1:** Gradient-Based Planning (GBP) via Gradient Descent

---

**Input:** Start state $z_1$, goal state $z_{\text{goal}}$, world model $f_\theta$, horizon $H$, optimization iterations $N$
**Output:** Optimal action sequence $\{\hat{a}_t\}_{t=1}^H$

Initialize action prediction $\{\hat{a}_t\}_{t=1}^H \sim \mathcal{N}(0, I_H)$
**for** $i = 1, \ldots, N$ **do**
    $\hat{z}_{H+1} \leftarrow \text{rollout}_f(z_1, \{\hat{a}_t\})_{H+1}$
    $\mathcal{L}_{\text{goal}} \leftarrow \| \hat{z}_{H+1} - z_{\text{goal}} \|_2^2$
    $\{\hat{a}_t\} \leftarrow \{\hat{a}_t\} - \eta \cdot \nabla_{\{\hat{a}_t\}} \mathcal{L}_{\text{goal}}$
**end**
**return** $\{\hat{a}_t\}_{t=1}^H$

---

As the planning objective is induced entirely by the world model, the success of GBP hinges on **(1)** the model accurately predicting future states under any candidate action sequence, and **(2)** the stability of this differentiable optimization. We present two training-time methods designed to improve these capabilities.

### 3.2 ONLINE WORLD MODELING

During GBP, the action sequences proposed by optimization frequently lie outside the distribution on which the world model was trained. The model is fit on a fixed dataset of expert trajectories, but GBP selects actions solely to drive predicted states toward a desired goal, without regard for whether those actions resemble expert behavior. This optimization process is known to induce adversarial inputs (Szegedy et al., 2013; Goodfellow et al., 2014), and in our setting produces out-of-distribution action sequences that lead the world model to make large prediction errors. Even when prediction errors are initially small, these actions push the model into unfamiliar regions of the state space, where errors compound over long-horizon planning.

To address this issue, we propose **Online World Modeling**, which iteratively corrects the trajectories produced by GBP and finetunes the world model on the resulting rollouts. Rather than training solely on expert demonstrations, we repeatedly incorporate trajectories induced by the planner itself, thereby expanding the region of latent states that the world model can reliably predict.

---

**Algorithm 2:** Online World Modeling

**Input:** Pretrained world model $f_\theta$, simulator dynamics function $h$, encoder $\Phi_\mu$, dataset of trajectories $\mathcal{T}$, online iterations $K$, horizon $H$, planning optimization iterations $N$

**Output:** Updated world model $f_\theta$

Initialize new trajectory dataset $\mathcal{T}'$

**for** $i = 1, \ldots, K$ **do**

    Sample trajectory $\tau_i = (z_1, a_1, z_2, a_2, \ldots, a_H, z_{H+1}) \sim \mathcal{T}$

    $\{\hat{a}_t\}_{t=1}^H \leftarrow \text{GBP}(z_1, z_{H+1}, p_\theta, H, N)$

    $\{s'_t\}_{t=2}^{H+1} \leftarrow \text{rollout}_h(s_1, \{\hat{a}_t\})$

    $\{z'_t\}_{t=2}^{H+1} \leftarrow \{\Phi_\mu(s'_t)\}_{t=2}^{H+1}$

    $\tau'_i \leftarrow (z_1, \hat{a}_1, z'_2, \hat{a}_2, \ldots, \hat{a}_H, z'_{H+1})$

    $\mathcal{T}' \leftarrow \mathcal{T}' \cup \tau'_i$

    Train $f_\theta$ on next-state prediction using $\mathcal{T}'$

**end**

**return** $f_\theta$

---

First, we conduct GBP using the first and goal latent states of an expert trajectory $\tau$, yielding a sequence of predicted actions $\{\hat{a}_t\}_{t=1}^H$. These actions might send the world model intro regions of latent state that lie outside of the training distribution. To correct for this, we obtain a *corrected trajectory*: the actual sequence of states that would result by executing the action sequence $\{\hat{a}_t\}_{t=1}^H$ in the environment using the true dynamics $h$ (in our setting, a simulator). The corrected trajectory,

$$\tau' = (z_1, \hat{a}_1, z'_2, \hat{a}_2, \ldots, z'_{H+1}) \tag{6}$$

is added to the dataset that the world model trains with every time the dataset is updated. Repeatedly training on the corrected trajectories ensures that the world model's behavior is adequately adjusted. We provide more detail in Algorithm 2 and illustrate our method in Figure 1.

Online world modeling is reminiscent of DAgger (Dataset Aggregation) (Ross et al., 2011), an online imitation learning method wherein a base policy network is iteratively trained on its own rollouts with the action predictions replaced by those from an expert policy. In a similar spirit, we invoke the ground-truth simulator as our expert world model that we imitate.

### 3.3 ADVERSARIAL WORLD MODELING

Rather than only updating the world model on trajectories encountered during planning, we propose an adversarial training method to train on latent states where the world model is anticipated to perform poorly, without conducting planning. These adversarial samples may lie outside the expert trajectory distribution that the world model was originally trained on, thereby promoting robustness to these regions of latent space during planning.

Adversarial training improves model robustness by optimizing performance under worst-case perturbations (Madry et al., 2018). An adversarial example is generated by applying a perturbation $\delta$ to

an input $x_i$ that maximally increases the model's loss. We have the following objective

$$\min_\theta \sum_i \max_{\delta \in \Delta} \mathcal{L}(f_\theta(x_i + \delta, y_i)), \tag{7}$$

where $\Delta = \{\delta : \|\delta\|_\infty \leq \epsilon\}$ to constrain the magnitude of perturbation. Training on these adversarially perturbed trajectories provides an alternative method to Online World Modeling of surfacing states that may be encountered during planning, without relying on GBP rollouts. Moreover, we find that it smooths the loss surface of the planning objective (see Figure 2), which in turn improves the stability of action-sequence optimization.

We generate adversarial latent states using the Fast Gradient Sign Method (FGSM) (Goodfellow et al., 2014), which efficiently approximates the worst-case perturbations that maximize prediction error (Wong et al., 2020). Although stronger iterative attacks such as Projected Gradient Descent (PGD) can be used, we find that FGSM delivers comparable improvements in GBP performance while being significantly more computationally efficient. This enables us to generate adversarial samples over entire large-scale offline imitation learning datasets. We denote this procedure **Adversarial World Modeling** and describe it below. See Appendix D for further design decisions.

---

**Algorithm 3:** Adversarial World Modeling

**Input:** Pretrained world model $f_\theta$, dataset of trajectories $\mathcal{T}$, action perturbation scaling $\lambda_a$, state perturbation scaling $\lambda_z$, horizon $H$, training iterations $N$, minibatch size $B$

**Output:** Updated world model $f_\theta$

Initialize new trajectory dataset $\mathcal{T}'$
**for** $i = 1, \ldots, N$ **do**
 Sample minibatch $\tau \leftarrow \{(z_1^j, a_1^j, z_2^j), (z_2^j, a_2^j, z_3^j), \ldots, (z_H^j, a_H^j, z_{H+1}^j)\}_{j=1}^B \sim \mathcal{T}$
 $(\epsilon_a, \epsilon_z) \leftarrow (\lambda_a \mathrm{std}(\{a_1^j, \ldots, a_H^j\}), \lambda_z \mathrm{std}(\{z_1^j, \ldots, z_{H+1}^j\}))$
 $(\alpha_a, \alpha_z) \leftarrow (1.25\epsilon_a, 1.25\epsilon_z)$
 $\delta_a \sim \mathrm{Uniform}(-\epsilon_a, \epsilon_a)$
 $\delta_z \sim \mathrm{Uniform}(-\epsilon_z, \epsilon_z)$
 **for** $t = 1, \ldots, H$ **do**
  $\nabla_{\delta_a}, \nabla_{\delta_z} \leftarrow \nabla_{\delta_a, \delta_z} \|f_\theta(z_t + \delta_z, a_t + \delta_a) - z_{t+1}\|_2^2$
  $\delta_a \leftarrow \mathrm{clip}(\delta_a + \alpha_a \mathrm{sign}(\nabla_{\delta_a}), -\epsilon_a, \epsilon_a)$
  $\delta_z \leftarrow \mathrm{clip}(\delta_z + \alpha_z \mathrm{sign}(\nabla_{\delta_z}), -\epsilon_z, \epsilon_z)$
  $a'_t \leftarrow a_t + \delta_a$
  $z'_t \leftarrow z_t + \delta_z$
 **end**
 $\tau' \leftarrow \{(z_1'^j, a_1'^j, z_2^j), (z_2'^j, a_2'^j, z_3^j), \ldots, (z_H'^j, a_H'^j, z_{H+1}^j)\}_{j=1}^B$
 Train $f_\theta$ on next-state prediction using $\tau'$.
**end**
**return** $f_\theta$

---

Let $\epsilon_a, \epsilon_z$ denote the radius of the perturbation to the actions $\{a_t\}$ and latent states $\{z_t\}$ respectively. For each state-action pair in a given minibatch, we look for small changes to the latent state or action that most increase the world model's prediction error. We compute gradients $\nabla_{\delta_a}, \nabla_{\delta_z} = \nabla_{\delta_a, \delta_z} \|f_\theta(z_t + \delta_z, a_t + \delta_a) - z_{t+1}\|_2^2$ and take a gradient ascent step (i.e., in the direction that worsens the prediction) with size $\alpha_a, \alpha_z = 1.25\epsilon_a, 1.25\epsilon_z$. We clip the result so that the perturbation stays within our radii. This procedure corresponds to a single FGSM/PGD-style adversarial update, producing perturbations that lie on the edge of the allowed region where they are maximally challenging for the model. See Algorithm 3 for the complete algorithm.

We perform grid-search on the scaling factors $\lambda_a, \lambda_z$ and find that AWM is robust for $0 \leq \lambda_a \leq 1$ and $0 \leq \lambda_z \leq 0.5$. Across experiments, we found that fixing our perturbation radii $\epsilon_a, \epsilon_z$ to the standard deviation of the initial minibatch is stable across all experiments. Updating this estimate for each batch yields as in Algorithm 3 yields no consistent improvement in final planning performance. See Appendix D.2 for details and results.

| | PushT | | | PointMaze | | | Wall | | |
|---|---|---|---|---|---|---|---|---|---|
| | GD | Adam | CEM | GD | Adam | CEM | GD | Adam | CEM |
| DINO-WM | 38 | 54 | 78 | 12 | 24 | 90 | 2 | 10 | 74* |
| + MPC | 56 | 76 | 92 | 42 | 68 | 90 | 12 | 80 | 82 |
| Online WM | 34 | 52 | 90 | 20 | 14 | 62 | 16 | 18 | 54* |
| + MPC | 50 | 76 | 92 | **54** | 88 | 96 | **38** | 80 | 90 |
| Adversarial WM | 56 | 82 | **94** | 32 | 70 | 88 | 32 | 34 | 30* |
| + MPC | **66** | **92** | 92 | 50 | **94** | **98** | 14 | **94** | **94** |

Table 1: **Planning Results.** We evaluate the planning performance of our finetuned world models against DINO-WM  (Zhou et al., 2025) on 3 tasks in terms of success rate (%) using both open-loop and model predictive control (MPC) procedures. For each task, we perform gradient-based planning using both stochastic gradient descent (GD) and Adam  (Kingma & Ba, 2014), and search-based planning using the cross-entropy method (CEM).

Adversarial World Modeling offers practical benefits relative to Online World Modeling. Unlike Online approaches, it requires no additional trajectory rollouts during training. This is a significant advantage in settings where simulation is expensive or infeasible. In addition, adversarial training has been shown to smooth the loss surface (Mejia et al., 2019), which can stabilize and simplify the planning optimization. We observe this effect empirically in Figure 2.

## 4 EXPERIMENTS

We evaluate the performance of our methods using pretrained world models from DINO World Model (Zhou et al., 2025) on 3 tasks: PushT, Point-Maze, and Wall. These tasks are adopted from DINO-WM. Specifically, we evaluate our method on the task of driving a system from an initial configuration $o_1$ to a target configuration $o_{\text{goal}}$, both specified as observations in $\mathcal{O}$. We report planning results, both in Open-Loop and in MPC, in Table 1. In the open-loop setup, we run Algorithm 1 with the initial state $\Phi_\mu(o'_1)$ and evaluate the predicted actions. In the MPC setup, we run Algorithm 1 once for each MPC step (using $\Phi_\mu(o'_1)$ as the initial state for the first MPC step), rollout the predicted actions $\{\hat{a}_t\}$ in the environment simulator to reach latent state $\hat{z}_{H+1}$, and set $\hat{z}_1 = \hat{z}_{H+1}$ for the next MPC iteration. We report all finetuning and planning optimization hyperparameters in Table 3.

We primarily adopt the latent world model framework from DINO World Model (DINO-WM) (Zhou et al., 2025) for its strong generalization across tasks. We also study the use of the IRIS (Micheli et al., 2023) world model architecture in Appendix B.3. For DINO-WM, the embedding function $\Phi_\mu$ is implemented using the pre-trained DINOv2 encoder introduced in (Oquab et al., 2024) and remains frozen while finetuning the transition model $f_\theta$. $f_\theta$ is implemented using the ViT architecture introduced in  (Dosovitskiy et al., 2021). We use a VQVAE decoder (van den Oord et al., 2018) to visualize latent states. To initialize the action sequence before planning, we evaluate randomly sampling from a standard normal distribution or sampling from an initialization network. For the initialization network, we train a function $g_\theta : \mathcal{Z} \times \mathcal{Z} \to \mathcal{A}^T$, such that $g_\theta(z_1, z_g) = \{\hat{a}_t\}_{t=1}^T$. We find that the randomly sampled initialization tends to yield greater planning performance. However, we analyze the impact of including this initialization network $g_\theta$ in Appendix B.1.

During planning, we set $\mathcal{L}_{\text{goal}}$ in Algorithm 1 to a *weighted goal loss* to obtain a gradient from each predicted state instead of simply the last one. We find empirically that this task assumption generalizes to both navigation (e.g., PointMaze and Wall) and non-navigation tasks (e.g., PushT), improving or matching performance of the final-state loss. We provide the exact formulation and more details in Appendix A.4. We additionally evaluate using the Adam optimizer (Kingma & Ba, 2014) during GBP. Although use of the Adam optimizer improves performance significantly over GD for DINO-WM and our finetuned world models, we find that this approach does not scale

---

* We could not reproduce the Wall environment open-loop CEM success rate reported in DINO-WM (74% over our 32%), so we report their better value.

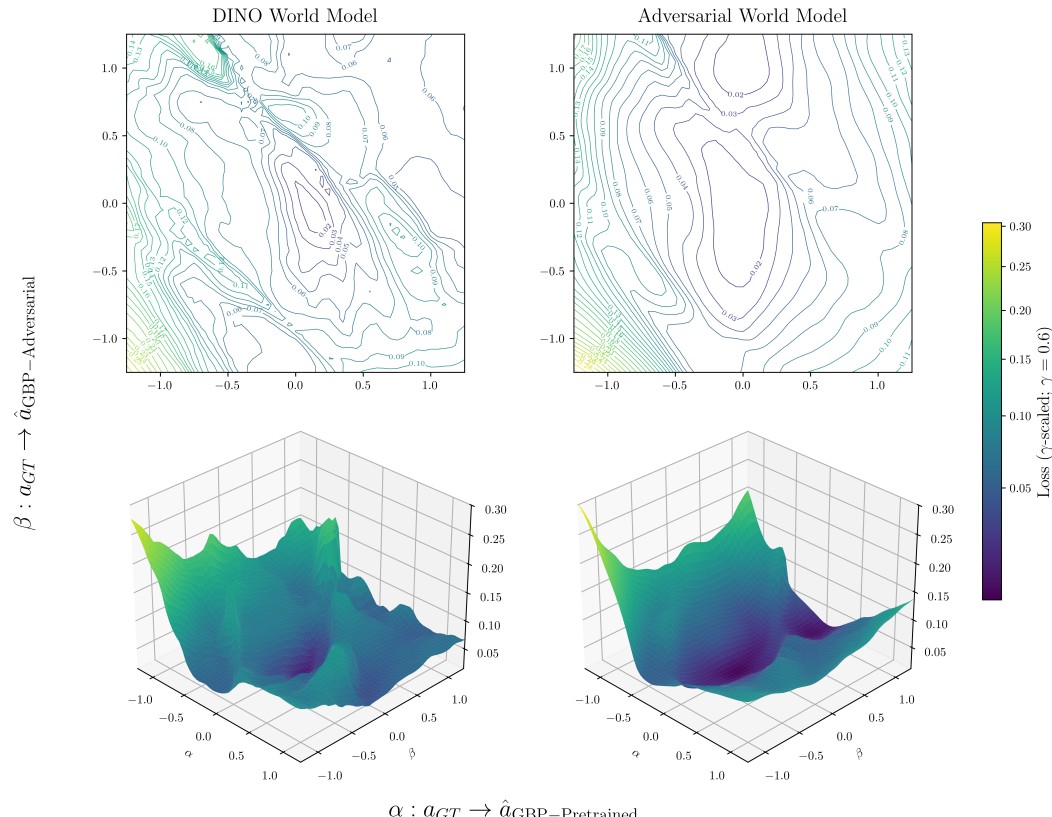

Figure 2: Optimization landscape of the DINO-WM (Zhou et al., 2025) before and after applying our Adversarial World Modeling objective on the Push-T task. Adversarial World Modeling yields a smoother and more convex landscape, with a broader basin around the optimum. Visualization details in Appendix C.

performance to match search-based methods. The optimization landscape of the pretrained DINO-WM is highly non-convex and the choice of optimizer alone fails to address this underlying problem.

## 4.1 PLANNING RESULTS

On all three tasks, we outperform Gradient-Based Planning with Gradient Descent and either match or outperform DINO-WM with the more expensive CEM. In the open-loop setting, we achieve +44% on Push-T, +58% on PointMaze, and +32% on Wall over Gradient Descent. In the MPC setting, GBP with Adversarial WM outperforms CEM with DINO-WM on PointMaze and Wall and matches CEM on PushT. We find that Adam almost always yields higher planning performance over Gradient Descent, and we hypothesize that this is due to its ability to traverse a more complex optimization landscape.

While both Online World Modeling and Adversarial World modeling bootstrap new data to improve the robustness of our world model at GBP-time, the distributions they induce are quite different. Whereas Online World Modeling anticipates and covers the distribution seen at planning time, Adversarial World Modeling exploits the current loss landscape of the world model to encourage local smoothness near expert trajectories. For most settings, we find that Adversarial World Modeling outperforms Online World Modeling, with the exception of PointMaze. We hypothesize this is due to the PointMaze task benefiting from search. Motivated by this result, we hypothesize that these two objectives balance different concerns — the former compounding errors over world model roll-outs (impacting the accuracy of predictions) and the latter in optimization (impacting the accuracy of actions between two states).

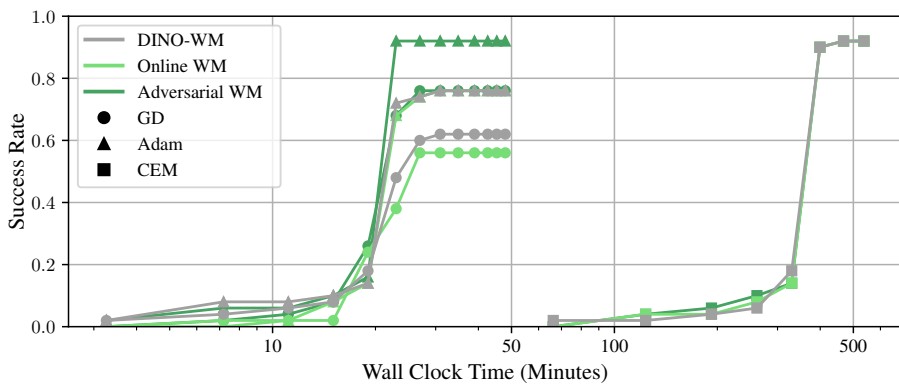

Figure 4: Planning efficiency of DINO-WM, Online WM, and Adversarial WM using both GBP methods and CEM on the PushT task.

To demonstrate the advantages of Adversarial World Modeling in more complex environments where the simulator may be very costly and the number of action dimensions is larger, we also evaluate planning performance on two robotic manipulation tasks in Appendix B.2.

### 4.2 TRAIN-TEST GAP

Comparing the world model error between training and planning trajectories allows us to evaluate if the world model will perform well during planning even if it is trained to convergence on expert trajectories. We evaluate world model error as the deviation between the world model's predicted next latent state, and the next latent state given by the environment simulator. Formally, given an initial state $s_1$ and a sequence of actions $\{a_t\}$ that either come from

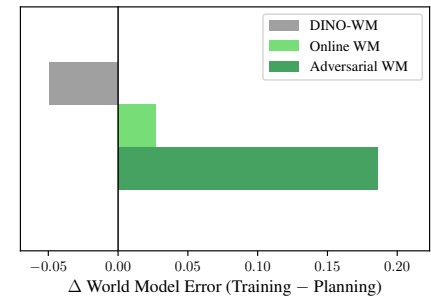

Figure 3: Difference in World Model Error between expert trajectories and planning trajectories on PushT.

the training dataset or from a planning procedure, the world model error at timestep $t$, $\Delta_t$, is given by,

$$\Delta_t = \|f_\theta(\Phi_\mu(s_t), a_t) - \Phi_\mu(h(s_t, a_t))\|^2, \quad s_{t+1} = h(s_t, a_t). \tag{8}$$

This error is averaged over all timesteps of a trajectory. If the difference between the world model error over the expert trajectories and over the planning trajectories is negative, then the world model will perform worse on sequences of actions produced during planning. Figure 3 demonstrates that this is the case with DINO-WM, but not with Online World Modeling and Adversarial World Modeling. Additional train-test gap results can be found in Appendix B.6.

### 4.3 PLANNING COMPUTATIONAL EFFICIENCY

When using a world model to conduct planning in real world environments, fast inference is crucial for actively interacting with an environment. On all three tasks, we find that GBP with Adversarial World Modeling is able to match or come near the best performing world model when planning with CEM, in over an order of magnitude less wall clock time. We compare wall clock times across world models and planning procedures for PushT in Figure 4. The planning efficiency results for PointMaze and Wall can be found in Appendix B.7.

## 5 CONCLUSION

In this work, we present Online World Modeling and Adversarial World Modeling as techniques to close the train-test gap that emerges between training world models on next-state prediction and

using an iterative planning process to take action. By demonstrating improved performance on GBP and occasionally surpassing CEM, we hope GBP can be more adopted for planning with world models. Future directions for this work are twofold. Firstly, we seek to extend our method to directly improve the quality of action gradients during planning. Secondly, evaluating the robustness of our methods to long-horizon planning will be crucial for practical usage.

**Limitations.** Our work assumes having access to offline datasets with representative state-action data, which may not be available for complex environments in which simulators are not provided. Likewise, our Online World Modeling algorithm relies on the presence of a computationally efficient environment simulator, which may be infeasible for real-world tasks.

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

# A EXPERIMENTAL DETAILS

## A.1 TASK DETAILS

**PushT:** This task introduced by Chi et al. (2024) uses an agent interacting with a T-shaped block to guide both the agent and block from a randomly initialized state to a feasible goal state within 25 steps. We use the dataset of 18500 trajectories given in Zhou et al. (2025), in which the green anchor serves purely as a visual reference. We draw a goal state from one of the noisy expert trajectories at 25 steps from the starting state.

**PointMaze:** In this task introduced by Fu et al. (2021), a force-actuated ball which can move in the $x, y$ Cartesian directions has to reach a target goal within a maze. We use the dataset of 2000 random trajectories present in Zhou et al. (2025), with a goal state chosen 25 steps from the starting state.

**Wall:** This task introduced by DINO-WM (Zhou et al., 2025) features a 2D navigation environment with two rooms separated by a wall with a door. The agent's task is to navigate from a randomized starting location in one room to a random goal state in the other room, passing through the door. We use the dataset of 1920 trajectories as provided in DINO-WM, with a goal state chosen 25 steps from the starting state.

**Rope:** In this task introduced by Zhang et al. (2024) a simulated Xarm must push a piece of rope into the goal orientation. We use the dataset of 1000 trajectories of 20 steps each provided in DINO-WM.

**Granular:** In this task introduced by Zhang et al. (2024) a simulated Xarm must push around one hundred small particles into the goal configuration. We use the dataset of 1000 trajectories of 20 steps each provided in DINO-WM.

We reproduce the dataset statistics used to train the base world model for each environment from Zhou et al. (2025). We use the same datasets for our alternative world model architecture ablation in Section B.3.

| Environment | H | Frameskip | Dataset Size | Trajectory Length |
|---|---|---|---|---|
| Push-T | 3 | 5 | 18500 | 100-300 |
| PointMaze | 3 | 5 | 2000 | 100 |
| Wall | 1 | 5 | 1920 | 50 |
| Rope | 1 | 1 | 1000 | 5 |
| Granular | 1 | 1 | 1000 | 5 |

Table 2: Trajectory datasets used to pretrain the base DINO-WM and IRIS world models.

## A.2 CEM ALGORITHM

We detail the cross-entropy method used in our planning experiments in Algorithm 4.

## A.3 FINETUNING AND PLANNING HYPERPARAMETERS

In Table 3, we list all shared hyperparameters used in training and planning.

We provide data quantity and synthetic data parameters for our Online and Adversarial World Modeling training setups in Table 5 and 4 respectively. In addition to the maintaining perturbation radii for the visual latent and action embeddings, we use a distinct radius for the proprioceptive embeddings. We empirically find that the scales of the visual and proprioceptive embeddings are incompatible and semantically distinct, thereby necessitating independent perturbation. Throughout all of our experiments, we set the perturbation radii of the action embedding and proprioceptive embedding identically for simplicity.

---

**Algorithm 4:** Cross-Entropy Method (CEM) Planning

---

**Input:** Current observation $o_0$, goal observation $o_g$, encoder $\Phi_\mu$, world model $f_\theta$,
horizon $H$, population size $N$, top-K selection $K$, iterations $I$
**Output:** Action sequence $\{\hat{a}_t\}_{t=1}^H$

$\hat{z}_1 \leftarrow \Phi_\mu(o_1)$
$z_g \leftarrow \Phi_\mu(o_g)$
Initialize Gaussian distribution parameters: mean $\mu_0$, covariance $\Sigma_0$
**for** $i = 1, \ldots, I$ **do**
    Sample $N$ action sequences $\{a_{1:H}^{(j)}\}_{j=1}^N \sim \mathcal{N}(\mu_{i-1}, \Sigma_{i-1})$
    **for** $j = 1, \ldots, N$ **do**
        $\hat{z}_1^{(j)} \leftarrow \hat{z}_1$
        **for** $t = 2, \ldots, H + 1$ **do**
            $\hat{z}_t^{(j)} \leftarrow f_\theta(\hat{z}_{t-1}^{(j)}, a_{t-1}^{(j)})$
        **end**
        $C^{(j)} \leftarrow \|\hat{z}_{H+1}^{(j)} - z_g\|^2$
    **end**
    Select $K$ sequences with lowest cost: $\mathcal{E} = \{a^{(j)}\}_{\text{top-}K}$
    $\mu_i \leftarrow \frac{1}{K} \sum_{a \in \mathcal{E}} a$
    $\Sigma_i \leftarrow \frac{1}{K} \sum_{a \in \mathcal{E}} (a - \mu_i)(a - \mu_i)^\top$
**end**
**return** $\mu_I$ as the final action sequence estimate $\{\hat{a}_t\}_{t=1}^H$

---

| Name | Value |
|------|-------|
| Image size | 224 |
| Optimizer | AdamW |
| Predictor LR | 1e-5 |

(a) Finetuning Parameters

| Name | GD | Adam |
|------|------|------|
| Opt. steps | 300 | 300 |
| LR | 1.0 | 0.3 |

(b) Open-Loop Planning

| Name | GD | Adam |
|------|------|------|
| MPC steps | 10 | 10 |
| Opt. steps | 100 | 100 |
| LR | 1 | 0.2 |

(c) MPC Parameters

Table 3: We report **(a)** shared hyperparameters for OWM/AWM finetuning across all environments, **(b)** open-loop planning optimization parameters, and **(c)** closed-loop (MPC) planning optimization parameters.

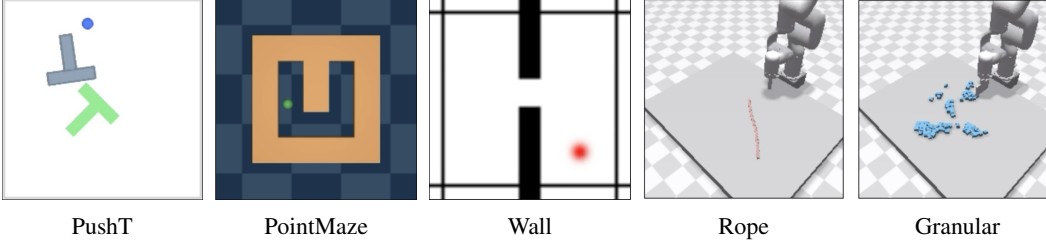

| PushT | PointMaze | Wall | Rope | Granular |

Figure 5: Illustrations of the three tasks used in our main experiments and the two robotic manipulation tasks we further study in Appendix B.2. Images from Zhou et al. (2025).

## A.4 WEIGHTED GOAL LOSS

To facilitate progress towards the goal in Gradient-based Planning, we introduce an alternate loss function: Weighted Goal Loss (WGL). Instead of the standard goal loss function that only minimizes the $\ell_2$-distance between the final latent state produced by planning actions and the goal latent state,

| Environment | # Rollouts | Batch Size | GPU | Epochs | $\epsilon_{\text{visual}}$ | $\epsilon_{\text{proprio}}$ | $\epsilon_{\text{action}}$ |
|---|---|---|---|---|---|---|---|
| PushT | 20000 (all) | 16 | 8x B200 | 1 | 0.05 | 0.02 | 0.02 |
| PointMaze | 2000 (all) | 16 | 1x B200 | 1 | 0.20 | 0.08 | 0.08 |
| Wall | 1920 (all) | 48 | 1x B200 | 2 | 0.20 | 0.08 | 0.08 |

Table 4: Training parameters for Adversarial World Modeling as reported in Table 1.

| Environment | # Rollouts | Batch Size | GPU | Epochs |
|---|---|---|---|---|
| PushT | 6000 | 32 | 4x B200 | 1 |
| PointMaze | 500 | 32 | 4x B200 | 1 |
| Wall | 1920 (all) | 80 | 4x B200 | 1 |

Table 5: Training parameters for Online World Modeling as reported in Table 1.

WGL encourages intermediate latent states to also be close to the goal latent state. Formally,

$$\mathcal{L}_{\text{WGL}} = \frac{1}{H} \sum_{i=2}^{H+1} w_i \|\hat{z}_i - z_{\text{goal}}\|_2^2. \tag{9}$$

where the sequence of normalized weights $\{w_i\}_2^{H+1}$ is a hyperparameter choice. Empirically, we find that using this objective for Gradient-Based Planning either maintains or improves planning performance. For PointMaze and Wall, we found that exponentially upweighting later states in the planning horizon improved planning performance, so we set $w_i = 2^i$. For PushT, we found that exponentially upweighting earlier states improved planning performance, so we set $w_i = (1/2)^i$. We leave the optimal selection of this sequence of weights as future work.

## B ADDITIONAL EXPERIMENT RESULTS

### B.1 INITIALIZATION NETWORK

Motivated by the hypothesis that the optimization landscape is rugged (see Figure 2 for some evidence of this), we train an initialization network $g_\theta : \mathcal{Z} \times \mathcal{Z} \to \mathcal{A}^T, g_\theta(z_1, z_g) = \{\hat{a}_t\}$ to initialize a sequence of actions for gradient-based planning. We provide details on training the initialization

---

**Algorithm 5:** Initialization Network Training

**Input:** Initialization network $g_\theta$, LR $\eta$, dataset of trajectories $\mathcal{T}$, iterations $N$, horizon $H$
**Output:** Trained initialization network $g_\theta$

**for** $i = 1, \ldots, N$ **do**
    Sample trajectory $\tau_i = (z_1, a_1, z_2, a_2, \ldots, a_H, z_{H+1}) \sim \mathcal{T}$
    $\{\hat{a}_t\}_{t=1}^{H} \leftarrow g_\theta(z_1, z_{H+1})$
    $\mathcal{L}_{\text{actions}} \leftarrow \sum_{t=1}^{H} \|\hat{a}_t - a_t\|_2^2$
    $\theta \leftarrow \theta - \eta \nabla_\theta \mathcal{L}_{\text{actions}}$
**end**
**return** $g_\theta$

---

network $g_\theta$ in Algorithm 5. We train $g_\theta$ on a single epoch over the trajectories in the task's training dataset.

We show results of including the initialization network in GBP for each task in Table 6. Comparing to Table 1, we see that for both GD and Adam, the initialization network only performs comparably in the PushT environment compared to a random initialization.

| | PushT | | PointMaze | | Wall | |
|---|---|---|---|---|---|---|
| | GD+IN | Ad+IN | GD+IN | Ad+IN | GD+IN | Ad+IN |
| DINO-WM | 44 | 62 | 16 | 14 | 4 | 12 |
| + MPC | 60 | 84 | 40 | 54 | 6 | 32 |
| Online WM | 56 | 66 | 8 | 28 | 10 | 18 |
| + MPC | 52 | 82 | 40 | 46 | 2 | 22 |
| Adversarial WM | **74** | **90** | 22 | 36 | 18 | 24 |
| + MPC | 74 | 90 | **44** | **56** | **24** | **48** |

Table 6: For both gradient descent (GD) and Adam (Ad), we evaluate initializing the actions for gradient-based planning (GBP) from the initialization network (IN) instead of a normal distribution.

| | Rope | | Granular | |
|---|---|---|---|---|
| | GD | CEM | GD | CEM |
| DINO-WM | 1.73 | 0.93 | 0.30 | **0.22** |
| Adversarial WM | **0.93** | **0.82** | **0.24** | 0.28 |

Table 7: Planning performance measured with Chamfer Distance (less is better) on two robotic manipulation tasks: Rope and Granular.

## B.2 ROBOTIC MANIPULATION TASKS

We evaluate Adversarial World Modeling on two robotic manipulation tasks: Rope and Granular. Planning results for both tasks can be found in Table 7. To measure the accuracy of planned actions, we evaluate the Chamfer distance between the goal set of keypoints and the predicted set of keypoints.

## B.3 DIFFERENT WORLD MODEL ARCHITECTURE

We ablate the use of the DINO-WM architecture by evaluating planning performance with the IRIS (Micheli et al., 2023) architecture. Specifically, IRIS uses a VQ-VAE (van den Oord et al., 2018) for both the encoder and decoder, and a standard decoder-only Transformer (Vaswani et al., 2017). We find that even with a learned encoder, Adversarial World Modeling improves GBP performance and even CEM performance. Planning success rates of the IRIS architecture for the Wall task are reported in Table 8.

| | GD | CEM |
|---|---|---|
| IRIS | 0 | 4 |
| IRIS + Online WM | 0 | 0 |
| IRIS + Adversarial WM | **8** | **6** |

Table 8: Planning results in terms of success rate using the IRIS (Micheli et al., 2023) architecture on the Wall Task.

## B.4 LONG HORIZON PLANNING

We evaluate GBP over a longer horizon in Table 9a. We use Adam in the MPC setting for each of these runs, setting a goal state 50 timesteps into the future drawn from an expert trajectory, a planning horizon of 50 steps, and 20 MPC iterations where we take a single action at each iteration. The dataset of held-out validation trajectories for the Wall environment does not contain expert trajectories of 50 timesteps, so we omit it from our evaluations. In comparison, our results in Table 1 use a goal state drawn 25 timesteps in the future and a planning horizon of 25 steps. We find that on the longer horizon, Adversarial World Modeling outperforms DINO-WM on PushT and both Adversarial and Online World Modeling outperform DINO-WM on PointMaze.

|  | PushT | PointMaze |
| --- | --- | --- |
| DINO-WM | 16 | 70 |
| Online WM | 16 | **96** |
| Adversarial WM | **26** | 88 |

(a) Long-Horizon GBP

|  | MPPI | GradCEM |
| --- | --- | --- |
| DINO-WM | 2 | 78 |
| Online WM | 2 | 74 |
| Adversarial WM | 2 | **84** |

(b) MPPI and GradCEM on PushT

Table 9: Performance for **(a)** long-horizon GBP and **(b)** the MPPI and GradCEM algorithms

### B.5 ADDITIONAL PLANNING ALGORITHMS

Additionally, we evaluate both the MPPI (Williams et al., 2017c) and GradCEM (Bharadhwaj et al., 2020b) algorithms under MPC on the PushT task in Table 9b. MPPI is an online, receding-horizon controller that samples and evaluates perturbed action sequences, executes the first action of the lowest-cost trajectory, and then replans from the updated state at each timestep.

GradCEM refines the candidate sequences used to update the estimated action distribution with gradient descent to provide a more accurate estimate of the true distribution's parameters. We see that Adversarial World Modeling outperforms DINO-WM with GradCEM. Additionally, GradCEM exhibits slightly lower performance than vanilla CEM. We hypothesize this is due to the memory requirements of gradient descent necessitating reducing the number of candidate sequences by a factor of 6 compared to vanilla CEM, leading to reduced accuracy in estimating the true action distribution.

For MPPI, we use 5 samples each MPC iteration, with 100 MPC steps. For GradCEM, we use 50 samples, 30 CEM steps, and 2 Adam steps per CEM step with an LR of 0.3. For GradCEM we take 10 MPC steps.

### B.6 ADDITIONAL TRAIN-TEST GAP RESULTS

We present additional results for the difference in World Model Error between training and planning for the PointMaze and Wall tasks in Figure 6. For both tasks, our methods have lower error during planning compared to training except for Online World Modeling on PointMaze, which is inconclusive due to the low magnitude of world model error. Planning actions are obtained after 300 steps of GBP with GD on 50 rollouts using the inital and goal state from a training trajectory.

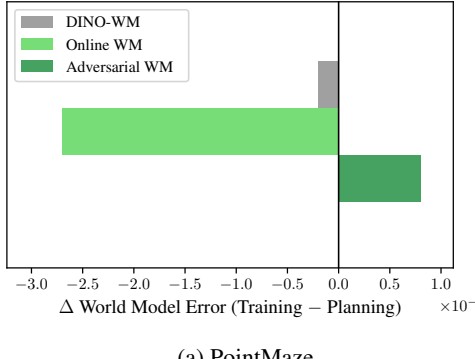
(a) PointMaze

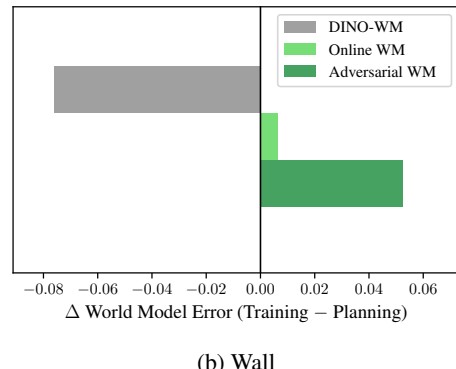
(b) Wall

Figure 6: Difference in World Model Error between expert trajectories and planning trajectories. Larger positive numbers indicate better performance on the actions seen during planning.

### B.7 PLANNING COMPUTATIONAL EFFICIENCY

For PointMaze and Wall, we compare the planning efficiency of DINO-WM and our two approaches across planning methodologies in Figures 7 and 8 respectively. We see that using Adam for GBP with our Adversarial WM achieves similar success rates to CEM in an order of magnitude less time.

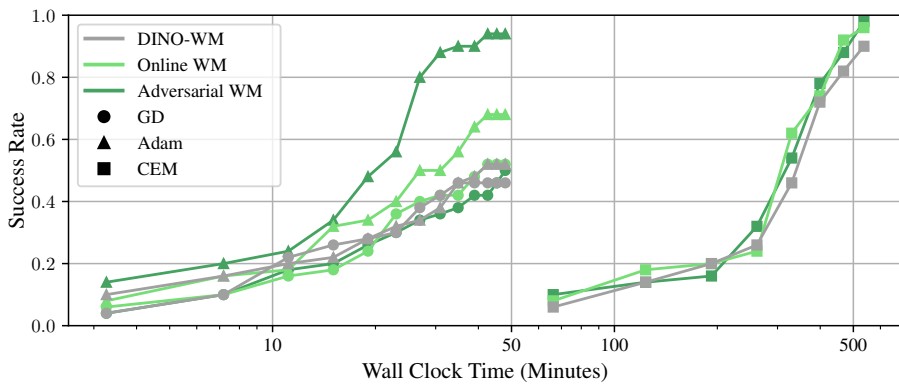

Figure 7: Planning efficiency of DINO-WM, Online WM, and Adversarial WM using both GBP methods and CEM on the PointMaze task.

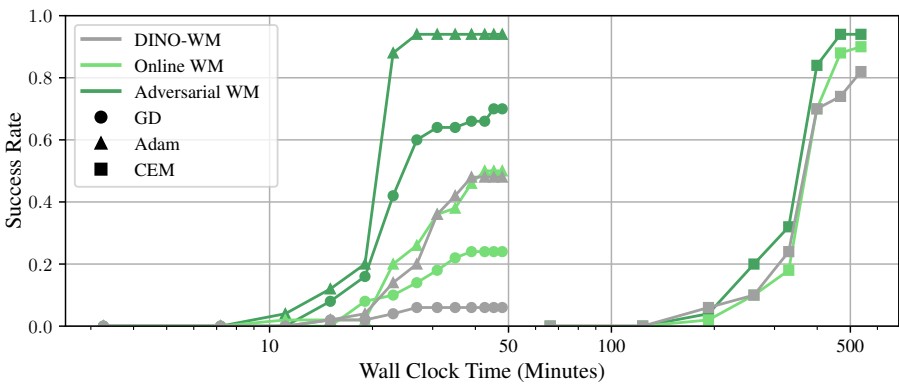

Figure 8: Planning efficiency of DINO-WM, Online WM, and Adversarial WM using both GBP methods and CEM on the Wall task.

### B.8 ROLLOUT INFERENCE TIME

To understand the additional cost of using the environment simulator in Online World Modeling, we record the wall clock time of rolling out 25 steps with the DINO-WM architecture and each environment simulator in Table 10. We see that in all environments, the simulator takes longer to rollout than the world model. We also note that the simulator for all 3 tasks is deterministic in terms of reproducing the training trajectories from their actions.

|  | PushT | PointMaze | Wall |
|---|---|---|---|
| Simulator | 0.959 | 0.717 | 4.465 |
| DINO-WM | 0.029 | 0.029 | 0.029 |

Table 10: Wall clock time (in seconds) of rolling out 25 steps with each environment simulator compared to the DINO-WM architecture.

## C VISUALIZING THE OPTIMIZATION LANDSCAPE

We visualize the loss landscape of both the DINO World Model before and after applying our Adversarial World Modeling objective. We perform a grid search over the subspace spanned by

1. $\hat{a}_{\text{GBP-Pretrained}}$: Gradient-Based Planning on original Dino World Model with 300 optimization steps of Adam with LR = 1e-3. We set a fixed initialization $a_{\text{init}}$.

2. $\hat{a}_{\text{GBP-Adversarial}}$: Gradient-Based Planning on our Adversarial World Model with 300 optimization steps of Adam with LR = 1e-3. We use the same $a_{\text{init}}$ as our initialization.

3. $a_{\text{GT}}$: the ground-truth actions from the expert demonstrator.

We define the axes as $\alpha = \hat{a}_{\text{GBP-Pretrained}} - a_{\text{GT}}$ and $\beta = \hat{a}_{\text{GBP-Adversarial}} - a_{\text{GT}}$, and compute the loss surface over a $50 \times 50$ grid spanning $\alpha, \beta \in [-1.25, 1.25]$.

## D   ADVERSARIAL WORLD MODELING: DESIGN DECISIONS

### D.1   FAST GRADIENT SIGN METHOD (FGSM) VS. PROJECTED GRADIENT DESCENT (PGD)

Projected Gradient Descent (PGD) has been used as an iterative method for generating adversarial perturbations (Madry et al., 2018). At each step, PGD takes a gradient ascent step and projects the result onto the space of allowed perturbations (some ball with radius $\epsilon$ around the input). Projection ($\Pi$) is typically via clipping or scaling. Formally,

$$\delta^{(k+1)} = \Pi_{\|\delta\|_{\infty} \leq \epsilon} \left( \delta^{(k)} + \alpha \cdot \nabla_x \mathcal{L}(f_\theta(x + \delta^{(k)}), y) \right) \tag{10}$$

However, this is computationally expensive to use for adversarial training as it requires an additional backward pass for each iteration. If one uses a single-step, replaces the gradient by its sign, and uses step size $\alpha = \epsilon$, this recovers the well-known Fast Gradient Sign Method (FGSM) update (Goodfellow et al., 2014).

$$\delta = \epsilon \, \text{sign} \left( \nabla_x \mathcal{L}(f_\theta(x), y) \right). \tag{11}$$

In Wong et al. (2020), the authors demonstrate that initializing $\delta$ in the $\ell_\infty$-ball with radius $\epsilon$ and performing FGSM adversarial training on these perturbations substantially improves robustness to PGD attacks and matches performance of PGD-based training. We leverage this observation to perform cheap adversarial training that only requires $2\times$ the backward passes of traditional supervised learning. In comparison, $K$-step PGD requires $K$ more backward passes ($3\times$ more for $K = 2$ and $4\times$ for $K = 3$). In Table 11, we show that 2/3-Step PGD does not consistently outperform FGSM, despite requiring a much larger training budget.

### D.2   SCALING FACTOR ($\lambda$) & PERTURBATION RADII ($\epsilon$) ABLATIONS

To assess the robustness of Adversarial World Modeling to the scaling factor and perturbation radius hyperparameters, we conduct an ablation study varying these two factors, shown in Figure 9. We evaluate $\lambda_a, \lambda_z \in [0.0, 0.02, 0.05, 0.20, 0.50, 1.0]^2$ and either fix $\epsilon_a, \epsilon_p, \epsilon_z$ to the standard deviation of the first minibatch ("Fixed") or recompute it for every minibatch ("Adaptive"). We observe no consistent improvement or degradation across any value of $\lambda_a$, for $0 \leq \lambda_z \leq 0.5$, or between the "Fixed" or "Adaptive" perturbation radii. We note that setting the visual scaling factor $\lambda_z$ too high (e.g., $0.5, 1.0$) can significantly degrade performance. We hypothesize that excessively large perturbations distort the semantic content of the visual latent state, pushing it outside the range of semantically equivalent representations.

| | | PointMaze | | | Wall | | |
|---|---|---|---|---|---|---|---|
| | Backward Passes | Min/Epoch | Open-Loop | MPC | Min/Epoch | Open-Loop | MPC |
| FGSM | **2** | **120** | 70 | 94 | **14** | **34** | **94** |
| 2-Step PGD | 3 | 165 | **80** | **96** | 20 | 8 | 90 |
| 3-Step PGD | 4 | 201 | 78 | 94 | 24 | 14 | 94 |

Table 11: Both Open-Loop and MPC (Closed-Loop) use the Adam optimizer with the same parameters as the main experiments.

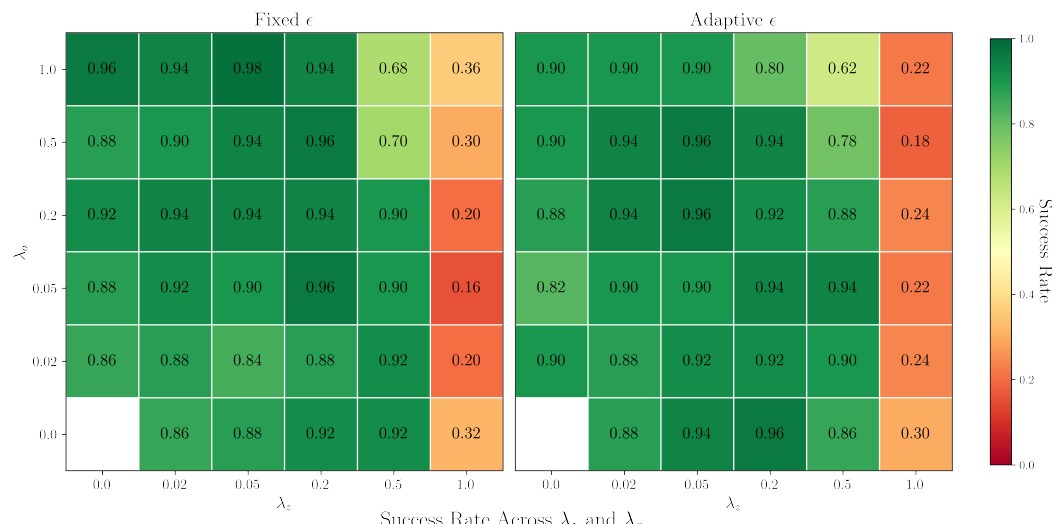

Figure 9: Success rate of closed-loop MPC planning using Adam on an Adversarial World Model trained with scaling factors $\lambda_a, \lambda_z$ and perturbation radii $\epsilon_a, \epsilon_z$ on the Wall environment. We that $0 \leq \lambda_z, \lambda_a \leq 0.2$ are stable for either "Fixed" or "Adaptive" perturbation radii.

# E    TRAJECTORY VISUALIZATION

We include visualizations of planning trajectories for DINO-WM, Online World Modeling, and Adversarial World Modeling to further study their success and failure modes. Visualizations for PushT and Wall can be found in Figures 10 and 11 respectively.

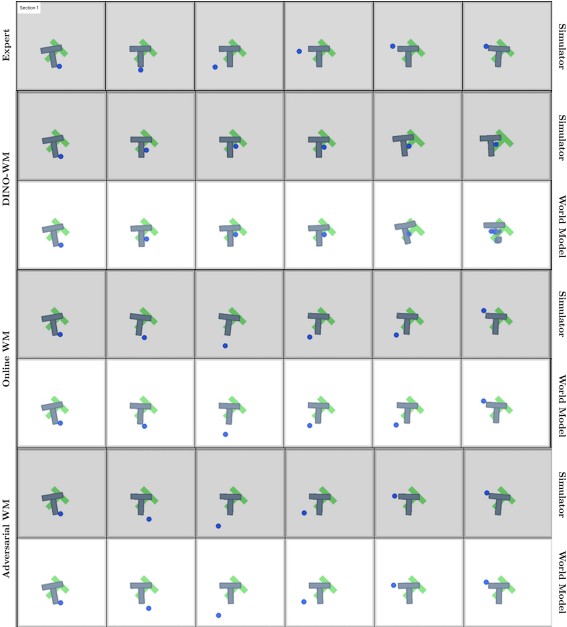

(a) We see that DINO-WM is more likely to enter states outside of the training distribution, and so the decoder is not able reconstruct the state accurately. This is not the case with Online World Modeling but it still fails to successfully reach the goal state. Adversarial World Modeling successfully completes the task.

(b) Again we notice the failure for DINO-WM's decoder to reconstruct states it encounters during planning, while this is not the case with Online World Modeling and Adversarial World Modeling, which both complete the task successfully.

Figure 10: Trajectory Visualizations of the PushT task. We plot the expert trajectory to reach the goal side, alongside both the simulator states and decoded latent states for DINO-WM, Online World Modeling, Adversarial World Modeling.

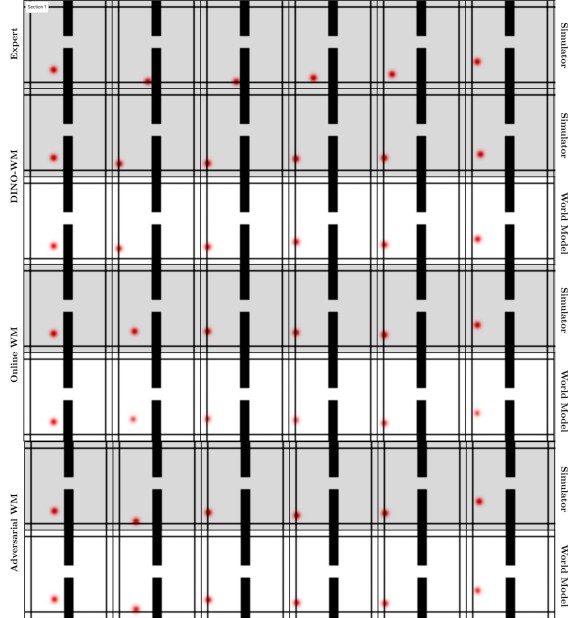

(a) In this challenging example, all three world models enter states through planning that their respective decoders cannot reconstruct, but only Online World Modeling is able to complete the task successfully.

(b) In this example, we see that DINO-WM predicts that it successfully completed the task according to its reconstructed last latent state, but the simulator indicates the true position to be off of the goal state. Online and Adversarial World Modeling correct for this and successfully complete the task.

Figure 11: Trajectory Visualizations of the Wall task. We plot the expert trajectory to reach the goal side, alongside both the simulator states and decoded latent states for DINO-WM, Online World Modeling, Adversarial World Modeling.

