# OpenReview forum: "Closing the Train-Test Gap in World Models for Gradient-Based Planning"
_ICLR.cc/2026/Conference — Submitted to ICLR 2026_

### Official Review · Reviewer_yvUu · 2025-10-31

**Soundness:** 2
**Presentation:** 2
**Contribution:** 1
**Rating:** 2
**Confidence:** 4

**Summary:**

This paper proposes a framework to improve gradient-based planning with learned world models by addressing the train-test gap. While existing world models achieve accurate next-state prediction, they often fail during long-horizon planning due to covariate shift and non-smooth loss landscapes, causing instability. The proposed solution introduces two training regimes built on a pretrained latent world model. Online World Modeling (OWM) mitigates distributional drift by iteratively collecting trajectories generated by the planner, executing them, and adding the resulting transitions into the training data, similar to DAgger. Adversarial World Modeling (AWM) improves local smoothness by training the model under perturbed latent states and actions, encouraging robustness to small input deviations. Using this refined world model, gradient-based planning directly optimizes action sequences in latent space toward a goal embedding. Experiments on three tasks demonstrate that both OWM and AWM reduce the simulator-model rollout discrepancy, enabling GBP to achieve higher success rates and faster convergence than standard teacher-forced training, and getting results similar to CEM while being significantly faster. However, the proposed regimes are conceptually simple and not novel, and are evaluated only on DINO-WM, so it is unclear whether the improvements generalize to other world-model architectures or tasks.

**Strengths:**

- Better results compared to baselines

- Two simple methods for dataset aggregation

**Weaknesses:**

- Novelty is very limited; the paper simply applies existing concepts (DAgger and adversarial training) into world model training and makes some further trivial changes (e.g., SGD to Adam optimizer)

- The method Adversarial World Modeling is not well-explained. In Algorithm 3, there are points that need to be clarified further. First, the trajectories that are added to the trajectory dataset are the original trajectories, not the ones that are updated. Even if the model is trained with the new trajectory (tao prime), the targets that we use $z_{t+1}'$ are also perturbed inputs/actions. The adversarial training should be done such that $f_\theta(a_t’, z_t’)$ is similar to $z_{t+1}$. In this version, it is $z_{t+1}’$. I believe training with a non-perturbed target is the right choice.

- Nearly half of the performance improvement seems to come from switching to Adam in planning, rather than the dataset augmentation methods that are the main contributions of this paper

- The details of the experimentation are missing: The way the initialization network is trained, details of the baseline CEM implementation, under which setup the data is collected for the graphs of Figure 3, what is meant by training $f_\theta$ on algorithm 2, 3 (do we take a batch or do 1 epoch), and architectural details of the networks. Some hyperparameter values (batch size, horizon length etc.) and hardware used are also not mentioned in the paper. Paragraph 4.3 is unclear, there should be more data about the used model

- Notational inconveniences:
  - (Mentioned above) Not generating and adding a new trajectory to the dataset in Algorithm 3.
  - There is no $p_\theta$ (line 257, 315) defined. I suppose it is $f_\theta$

- The only world model used is DINO-WM; it is unclear whether the proposed training regime would improve performance that much in other world models. In addition, the only non-gradient-based planner is CEM; MPPI or other planners could also be tested.

- It would also be a plus for the paper, if they have included more qualitative analysis for justifying the practical concept, including some scenes from their environments or having failure cases alongside successful trajectories.

- The number of tasks used in experimentation is three. It would be better to test the proposed methods on diverse tasks from potentially different frameworks/environments.

**Questions:**

- Elaborate what exactly is being compared in Figure 3. Which optimizer is used for each curve? Further discussion of these results is also needed; e.g. why does teacher forcing start with a significantly higher model loss difference?

- Instead of mentioning some hyperparameter values in the text, please include a single table with all hyperparameters that are used in your setup.

- In Algorithm 3, because of the way lambda parameters are chosen, don't we have clipping on most of the iterations? When this happens, does applying a perturbation in the direction of the gradient mean anything? Did you try not doing a gradient step and observe the results with just clipping applied? I suspect the results won't differ much.

- Some qualitative analysis (e.g. with example expert and generated trajectories) would strengthen your claims

---

> ### Author Response · Authors · 2025-11-27
> **Response to Reviewer yvUu - Part 1**
>
> Thank you for your time and feedback! We appreciate that you recognize the simplicity of our approaches along with their evidence for alleviating issues with Gradient-Based Planning and the empirical performance gains. We want to point out that since submission, we have made changes that cause AWM to be competitive or surpass CEM. We detail numerous new experiments, further analysis, and additional clarifiactions addressing your points below.
>
> **W1 ("...simply applies existing concepts..."):** To the best of our knowledge, we are the first to propose procedures specifically aimed at improving a world model’s ability to support Gradient-Based Planning (GBP). Despite its computational efficiency, GBP has historically exhibited poor empirical performance, leading prior work to favor sampling-based planners such as CEM. In contrast, our proposed methods enable GBP to consistently match or surpass CEM while achieving over 10× lower wall-clock time (see revised Table 1 and Figure 4). We attribute this substantial improvement to identifying and addressing two core hypotheses underlying GBP’s poor performance. Motivated by your feedback, we have significantly clarified these hypotheses in the revised Introduction, and we now provide stronger empirical support for both. First, we show that Online World Modeling and Adversarial World Modeling reduce the train–test gap in world modeling error, directly improving the accuracy of multi-step rollouts used for planning (revised Section 4.2, Figure 2). Second, we present new visualizations of the action-space loss surface optimized by GBP, demonstrating that Adversarial World Modeling produces a markedly smoother landscape, making gradient-based optimization far more tractable (revised Figure 2). In formulating these methods and studying their effect on GBP performance, we contribute to the understanding of GBP as a whole, a procedure that is often overlooked in the field despite its potential for greatly improving the computational efficiency of planning with world models.
>
> We want to emphasize that the methods we implement were originally developed for other areas, like image classification. We novelly apply techniques that were not intended to be used on world models, namely imitation learning and adversarial training, for the unique approach of finetuning world models to improve GBP performance. We believe that simplicity is a strength, and engineering complex new algorithms is unnecessary when re-purposing simple existing algorithms works well.
>
> **W2 ("...Adversarial World Modeling is not well explained..."):** We extend gratitude to the reviewer for reviewing the technical details. We agree that it is most intuitive for only the input state-action pair to be perturbed, not the next-state prediction, and this is exactly what we computed in our implementation of AWM. We have carefully re-written the formulation to clarify this fact, corrected the inputs and targets, and improved the motivating context surrounding all algorithms.
>
> **W3 ("...nearly half of the performance improvement seems to come from switching to Adam in planning..."):** It is true that Adam boosts performance over GD for DINO-WM, and we have clarified this further in our updated draft, however our techniques further boost performance and are required in order to close the gap with the more expensive CEM. Additionally, we have now further tested out a new weighted goal loss as the GBP objective, and found that it improved or maintained planning performance across all tasks. In short, this weighted goal loss also evaluates the distance of intermediate predicted states to the goal, weighted by a sequence of chosen weights. We include details of this loss function in Appendix A.4. With this new GBP objective we find that our AWM method is now competitive with or even outperforms CEM on the PointMaze and Wall environments as well as the PushT task. We have updated Table 1 in our submission with the latest evaluations and hope this reflects our method's broad improvement.
>
> **W4 ("...details of the experimentation are missing..."):** We have revised our submission with all of these details and have carefully included all other experimentation details as well: the initialization network (Appendix B.1), baseline CEM implementation (Appendix A.2), data collection (Appendix A.1), training details (A.3), and architectural details (Section 4).
>
> **W5 (Notational Inconveniences):** Thank you for pointing these out. We have fixed the notation accordingly and double-checked all our algorithms for notational correctness.

---

> ### Author Response · Authors · 2025-11-27
> **Response to Reviewer yvUu - Part 2**
>
> **W6 (Other architecture and planners):** We ablate the use of DINO-WM as the base world model architecture by using the IRIS [1] architecture instead. IRIS uses a VQ-VAE encoder and a standard decoder-only transformer for the predictor. We evaluate our methods using the IRIS architecture on the Wall task and find that Adversarial World Modeling makes improvements over base GBP and CEM. Success rate results are below and we have included them in Appendix B.3.
>
> | Method         | GD | CEM |
> |----------------|----|-----|
> | IRIS       | 0  | 4   |
> | IRIS + Online WM      | 0  | 0   |
> | IRIS + Adversarial WM | **8** | **6** |
>
> We evaluate the Model Predictive Path Integral (MPPI) planning method using an implementation based on [PyTorch-MPPI](https://github.com/UM-ARM-Lab/pytorch_mppi). MPPI is an online, receding-horizon controller that samples and evaluates perturbed action sequences, executes the first action of the lowest-cost trajectory, and then replans from the updated state at each timestep. We share results for PushT below and in Table 9b of the updated submission. Additional details for this experiment can be found in Appendix B.5. We also evaluate the hybrid GD + CEM algorithm GradCEM [2] under MPC. This method refines the candidate sequences used to update the estimated action distribution with gradient descent to provide a more accurate estimate of the true distribution's parameters. We share results for GradCEM on PushT below and in Table 9b of the updated submission, finding that Adversarial World Modeling outperforms DINO-WM. Additional details for this experiment can be found in Appendix B.5.
>
> |                  | MPPI | GradCEM |
> |------------------|------|---------|
> | DINO-WM          | 2    | 78      |
> | Online WM        | 2    | 74      |
> | Adversarial WM   | 2    | **84**  |
>
> **W7 ("It would also be a plus for the paper, if they have included more qualitative analysis..."):** We have added example trajectories and provide more detail in Q4 below.
>
> **W8 (Limited number of tasks):** We evaluate on two robotic manipulation tasks and see that Adversarial World Modeling significantly increases the performance of gradient-based planning and either matches or comes close to the base CEM performance. The tasks are called Rope and Granular, where a simulated XArm is tasked with pushing a piece of rope and a group of one hundred small cubes, respectively. These tasks are more challenging due to a larger number of action dimensions and the finer manipulation required. Furthermore, due to the large cost of the simulator, we only evaluate Adversarial World Modeling on these tasks. We present planning results in terms of Chamfer Distance from the goal set of keypoints (lower is better) in the table below. We have now included these results in Appendix B.2 of the revised submission.
>
> | Method          | Rope GD | Rope CEM |  Granular GD | Granular CEM  |
> |-----------------|---------|----------|--------------|---------------|
> | DINO-WM         | 1.73    | 0.93     | 0.30         | **0.22**      |
> | Adversarial WM  | **0.93**| **0.82** | **0.24**     | 0.28          |
>
> We also evaluate GBP over longer horizons of 50 steps. For PushT and PointMaze, we see that Adversarial World Modeling outperforms DINO-WM on PushT and PointMaze while Online World Modeling outperforms DINO-WM on PointMaze. We present success rate results for long-horizon planning below. Additional details for this experiment can be found in Appendix B.4.
>
> | Long Horizon          | PushT | PointMaze |
> |-----------------|---------|----------|
> | DINO-WM         | 16    | 70     |
> | Online WM         | 16 | **96**|
> | Adversarial WM  | **26**| 88 |
>
> **Q1 (Figure 3 Elaboration):** We have improved the clarity of the distribution-shift plot of Figure 3 in the revised submission. To analyze the gap between world modeling performance in training versus planning, we report the difference in teacher forcing loss between training trajectories and those obtained from planning. It is now clearer that our methods actually reverse the "train-test gap", by making the world model error lower on planning trajectories, and this is consistent across tasks. We have included further details of this analysis as you requested in Appendix B.6.
>
> **Q2 (Hyperparameter Values):** We have consolidated all hyperparameters for finetuning and planning across all methods and evaluation setups in Appendix A.3. We also describe various other statistics, such as dataset details, in tabular form in that section.

---

> > ### Author Response · Authors · 2025-11-27
> > **Response to Reviewer yvUu - Part 3**
> >
> > **Q3 (Clipping in AWM):** When clipping occurs, the direction of the gradient is still crucial as a $d$-dimensional perturbation always has at least $2^d$ possible configurations (either $-\epsilon$ or $\epsilon$ in the worst case). We conducted an experiment to validate this empirically. Following your suggestion, we do not do a gradient step and only apply clipping. On the Wall environment, with the same training compute as our main experiments, we achieve 82\% Success Rate in the closed-loop setting compared to 94\% with the gradient step. This performance is comparable to the pretrained model's performance of 80\%. We did, however, find that clipping/projection empirically occurs very often. We will add an aside in the main paper to motivate this.
> >
> > **Q4 (Example Trajectories):** We have included multiple qualitative trajectories produced by planning on the PushT and Wall tasks showing the expert trajectories and corresponding world model and ground-truth simulator states for pretrained DINO-WM, our Online World Model, and our Adversarial World Model in Appendix E. We cover cases where each model fails or succeeds and describe intuition for the cause.
> >
> > --------
> >
> > We believe your feedback has significantly improved our updated draft. We have conducted extensive experiments and made numerous changes to our draft based on your feedback, and we would appreciate it if you would consider raising your score in light of our response. Please let us know if you have additional questions we can address.
> >
> > [1] Micheli, V. et al. (2023). "Transformers are Sample-Efficient World Models." International Conference on Learning Representations 2023.
> >
> > [2] Bharadhwaj, H. et al. "Model-Predictive Control via Cross-Entropy and Gradient-Based Optimization." Conference on Learning for Dynamics and Control 2020.

---

### Official Review · Reviewer_P9Wi · 2025-11-01

**Soundness:** 2
**Presentation:** 2
**Contribution:** 1
**Rating:** 2
**Confidence:** 4

**Summary:**

The paper addresses the train–test gap between world-model training and test-time planning. World models are usually trained for one-step next-state prediction, but during gradient-based planning (GBP) they are unrolled for many steps, causing compounding errors and off-distribution states.
The authors propose two methods to mitigate this:
1. Online World Modeling (OWM), a DAgger-style approach: after planning, the predicted action sequence is executed in the true simulator to obtain corrected next states, which are added to the training set to update the world model.
2. Adversarial World Modeling (AWM): a latent-space FGSM-style attack. Perturb latent states and actions in the direction of largest prediction error and retrain on these adversarial samples.

Experiments on small 2-D tasks show moderate success-rate gains for GBP and some improvement over the Cross-Entropy Method at a fraction of its compute time.

**Strengths:**

- The “train–test gap” framing is intuitive and addresses a real limitation of current world-model planning.
- Both OWM and AWM are straightforward, well-explained extensions of DAgger and adversarial training to latent dynamics.
- Demonstrates that smoother, more robust dynamics enable faster differentiable planning.

**Weaknesses:**

- **Limited experimental scope:** evaluated only on a limited number of toy 2-D domains; no high-dimensional or real-robot settings, despite claims about scalability.
- **Incremental novelty:** both methods are direct adaptations of known techniques; no new architectural or theoretical contribution.
- **Missing baselines:** lacks comparison with modern hybrid planners (e.g. **TD-MPC2**).
- **Unclear wall-time reporting:** runtime results only shown for one task.
- **Writing clarity:** several explanations are confusing and could be better structured. Some citations seem off.
- **Limited generality:** OWM assumes simulator reset access which are impractical in real environments.

**Questions:**

1. How does AWM differ works that use adversarial perturbations to diversify state visitation?. You mention "In a similar spirit to our approach, (Zhang et al., 2025) introduce an adversarial attack method to encourage diverse state visitation distribution in a model-based RL setting." but that points to a survey paper.
2. Why report wall-clock efficiency only for PushT? Do other tasks show similar ratios vs. CEM?
3. If the simulator (h) is available (as required by OWM), why not plan directly in it instead of through a learned world model?
4. Could the approach extend to offline-only settings without simulator resets?

---

> ### Author Response · Authors · 2025-11-27
> **Response to Reviewer P9Wi - Part 1**
>
> Thank you for your time and feedback! We appreciate that you recognize our problem framing and methods to be intuitive. We note that in addition to this response post, we've also made a separate general post, with several clarifications and new results inspired by your comments. We detail numerous new experiments demonstrating an improvement in the competitiveness of our methods, further analysis, and additional clarifiactions addressing your points below.
>
> **W1 ("...evaluated only on a limited number of toy 2-D domains..."):**
> We evaluate on two robotic manipulation tasks and see that Adversarial World Modeling significantly increases the performance of gradient-based planning and either matches or comes close to the base CEM performance. The tasks are called Rope and Granular, where a simulated XArm is tasked with pushing a piece of rope and a group of one hundred small cubes, respectively. These tasks are more challenging due to a larger number of action dimensions and the finer manipulation required. Furthermore, due to the large cost of the simulator, we only evaluate Adversarial World Modeling on these tasks. We present planning results in terms of Chamfer Distance from the goal set of keypoints (lower is better) in the table below. We have now included these results in Appendix B.2 of the revised submission.
>
> | Method          | Rope GD | Rope CEM |  Granular GD | Granular CEM  |
> |-----------------|---------|----------|--------------|---------------|
> | DINO-WM         | 1.73    | 0.93     | 0.30         | **0.22**      |
> | Adversarial WM  | **0.93**| **0.82** | **0.24**     | 0.28          |
>
>
> We also evaluate GBP over longer horizons of 50 steps. We present success rate results for long-horizon planning below. For PushT and PointMaze, we see that Adversarial World Modeling outperforms DINO-WM on PushT and PointMaze while Online World Modeling outperforms DINO-WM on PointMaze.Additional details for this experiment can be found in Appendix B.4.
>
> | Long Horizon          | PushT | PointMaze |
> |-----------------|---------|----------|
> | DINO-WM         | 16    | 70     |
> | Online WM         | 16 | **96**|
> | Adversarial WM  | **26**| 88 |
>
> **W2 ("...both methods are direct adaptations of known techniques..."):**
> To the best of our knowledge, we are the first to propose the use of these procedures specifically aimed at improving a world model’s ability to support Gradient-Based Planning (GBP). Despite its computational efficiency, GBP has historically exhibited poor empirical performance, leading prior work to favor sampling-based planners such as CEM. In contrast, our proposed methods enable GBP to consistently match or surpass CEM while achieving over 10× lower wall-clock time (see revised Table 1 and Figure 4). We attribute this substantial improvement to identifying and addressing two core hypotheses underlying GBP’s poor performance. Motivated by your comments, we have significantly clarified these hypotheses in the revised Introduction, and we now provide stronger empirical support for both. First, we show that Online World Modeling and Adversarial World Modeling reduce the train–test gap in world modeling error, directly improving the accuracy of multi-step rollouts used for planning (revised Section 4.2, Figure 3). Second, we present new visualizations of the action-space loss surface optimized by GBP, demonstrating that Adversarial World Modeling produces a markedly smoother landscape, making gradient-based optimization far more tractable (revised Figure 2). In formulating these methods and studying their effect on GBP performance, we contribute to the understanding of GBP as a whole, a procedure that is often overlooked in the field despite its potential for greatly improving the computational efficiency of planning with world models.
>
> We want to emphasize that the methods we implement were originally developed for other areas, like image classification. We novelly apply techniques that were not intended to be used on world models, namely imitation learning and adversarial training, for the unique approach of finetuning world models to improve GBP performance. We believe that simplicity is a strength, and engineering complex new algorithms is unnecessary when re-purposing simple existing algorithms works well.

---

> > ### Author Response · Authors · 2025-11-27
> > **Response to Reviewer P9Wi - Part 2**
> >
> > **W3 ("...lacks comparison with modern hybrid planners (e.g. TD-MPC2)..."):** Below we provide a comparison between TD-MPC2 [1], DINO-WM, and Adversarial World Modeling on CEM+MPC across 5 tasks. The results of TD-MPC2 are obtained from the DINO-WM paper.
> >
> > | Model          | PushT (SR ↑) | PointMaze (SR ↑) | Wall (SR ↑) | Rope (CD ↓) | Granular (CD ↓) |
> > |----------------|--------------|------------------|-------------|-------------|------------------|
> > | TD-MPC2        | 0.00         | 0.00             | 0.00        | 2.52        | 1.21             |
> > | DINO-WM        | 0.78         | 0.90             | 0.32        | 0.93        | 0.22             |
> > | Adversarial WM | 0.94         | 0.88             | 0.30        | 0.82        | 0.28             |
> >
> >
> > Evidently, our chosen tasks are too challenging for TD-MPC2 to be a reasonable comparison, even with CEM which is typically more performative. Below, we also train and evaluate the IRIS [2] world modeling architecture on Online World Modeling and Adversarial World Modeling for the Wall task. We find that IRIS + Adversarial World Modeling makes improvements over the base IRIS model in GBP and CEM.
> >
> > | Method              | GD | CEM |
> > |---------------------|----|-----|
> > | IRIS                | 0  | 4   |
> > | IRIS + Online WM      | 0  | 0   |
> > | IRIS + Adversarial WM | **8** | **6** |
> >
> >
> >
> > **W4 ("...runtime results only shown for one task"):** We have included Wall-Clock time results for both PointMaze and Wall in Appendix B.7 as well as updated the Wall-Clock time plot for PushT to show the planning procedures to completion. It becomes evident that are methods with GBP are able to match the performance of CEM in 10x less wall clock time.
> >
> > **W5 ("...explanations are confusing..."):** We have improved clarification for some of our evaluation procedures in the revised submission, including a thorough explanation of the CEM Algorithm in Appendix A.2. We welcome your continued feedback on which explanations require further clarification. We have double-checked the accuracy of our citations and found no inaccuracies. Moreover, the citation you point out as incorrect was not a survey paper in our original submission. Please reference our response to Q1 below. If there are instances of incorrect citations we will promptly correct them.
> >
> > **W6 ("...OWM assumes simulator reset access..."):** Although Online World Modeling requires a simulator, Adversarial World Modeling directly perturbs offline trajectories without the use of a simulator. Furthermore, we find that Adversarial World Modeling significantly improves GBP performance in robotic manipulation tasks which demonstrates its applicability to real world settings.
> >
> > **Q1 (Existing Perturbation Work):** Compared to Selective STate-Aware Reinforcement adversarial attack (STAR) [3], Adversarial World Modeling differs in the "victim" of the adversarial perturbations and the goal of training on these perturbations. STAR produces adversarial trajectories for a policy network, while our trajectories are perturbed with respect to the next-state prediction loss of a world model. Furthermore, the sole goal of STAR is to robustify a policy to environmental perturbations, while an additional goal of Adversarial World Modeling is to smoothen the world model's loss surface for improved Gradient-Based Planning. We performed a visualization of the loss surface to confirm this effect in Figure 2. The citation correctly points to the citation for Zhang et al. 2025 ("State-aware perturbation optimization for robust deep reinforcement learning") in our original submission.
> >
> > **Q2 (Wall-Clock Results):** Since the world model architecture is kept constant across tasks, we observe the same difference in wall clock time between GBP and CEM (10x faster) for all three tasks that we run our main evaluations on. The Wall-Clock time results for both PointMaze and Wall have been included in Appendix B.7.
> >
> > **Q3 (Planning in the Simulator):** Simulators tend to be computationally expensive to run, especially as the environment becomes more complex. An accurate world model can perform planning much faster than a simulator. In fact, for all three tasks that we mainly evaluate on, we observe that the world model performs a rollout in much less wall clock time compared to the simulator for each environment (see Appendix B.8). Furthermore, most simulators are not differentiable, preventing the use of Gradient-Based Planning which we have noted to be much more computationally efficient than non-gradient-based CEM.
> >
> > **Q4 (Extension to Offline-Only):** One of the primary benefits of Adversarial World Modeling is that it does not require access to a simulator at all and can be trained with a purely offline dataset. We find that Adversarial World Modeling performs well in the robotic manipulation tasks (detailed in the general response) where obtaining online trajectories is expensive due to a costly simulator.

---

> > > ### Author Response · Authors · 2025-11-27
> > > **Response to Reviewer P9Wi - Part 3**
> > >
> > > We believe your feedback has significantly improved our updated draft. We have conducted extensive experiments and made numerous changes to our draft based on your feedback, and we would appreciate it if you would consider raising your score in light of our response. Please let us know if you have additional questions we can address.
> > >
> > > [1] Hansen, N. et al. "TD-MPC2: Scalable, Robust World Models for Continuous Control." International Conference on Learning Representations 2024.
> > >
> > > [2] Micheli, V. et al. (2023). "Transformers are Sample-Efficient World Models." International Conference on Learning Representations 2023.
> > >
> > > [3] Zhang, Z et al. "State-Aware Perturbation Optimization for Robust Deep Reinforcement Learning." arXiv:2503.20613, 2025.

---

### Official Review · Reviewer_rdsc · 2025-11-01

**Soundness:** 3
**Presentation:** 3
**Contribution:** 3
**Rating:** 6
**Confidence:** 3

**Summary:**

The paper studies why gradient based planning with learned world models underperforms strong search based planners and proposes two training time data augmentation strategies to close the resulting train test gap between next state prediction training and action sequence optimization at test time.
The setting uses a latent world model built on a frozen DINOv2 encoder and a ViT transition model trained with teacher forcing on offline expert trajectories. At test time, actions are optimized by backpropagating through the world model to minimize latent distance to a goal. The authors observe that gradient based planning drives the model into out of distribution latent regions and exploits modeling errors. They introduce two finetuning procedures that expand the effective training distribution to better match the distributions induced by planning:
- Online World Modeling: run gradient based planning between the first and last latent states of expert trajectories, execute the planned actions in a trusted simulator to obtain corrected rollouts, and aggregate these corrected trajectories to further train the transition model.
- Adversarial World Modeling: without running a simulator, generate adversarial perturbations in latent and action space around training minibatches using a one step FGSM style update with adaptive radii and clip the perturbations to create hard examples for robust finetuning.

**Strengths:**

- Clear identification of a practical train test gap for world models used with gradient based planning and a concrete diagnosis that planning induces out of distribution trajectories and adversarial solutions in latent space.
- Two simple and implementable finetuning strategies that are compatible with existing latent world models. Online World Modeling leverages a simulator to correct planned rollouts. Adversarial World Modeling promotes robustness with lightweight one step perturbations in latent and action space.
- Empirical improvements across three tasks from the DINO World Model suite. The paper reports sizable success rate gains over a teacher forcing baseline for both open loop and MPC settings and shows that gradient based planning with Adam benefits most from the proposed finetuning.
- Evidence that the gap between simulator and world model rollouts during planning is reduced, which directly supports the paper’s motivation to align training and test time distributions.

**Weaknesses:**

- Limited experimental breadth. Only three simulated tasks are considered and all come from the same DINO World Model setup. The work would be more convincing with additional domains or harder long horizon tasks.
- Comparisons to alternative strong planning baselines are incomplete. Prior hybrid methods that interleave CEM and gradient steps are not compared. There is no comparison to iLQR or DDP on lower dimensional variants, and MPPI is not included despite being common in this setting.
- The adversarial finetuning design uses heuristic choices such as adaptive batch standard deviation radii, fixed scaling factors, and one step updates. The paper lacks ablations that justify these choices or explore sensitivity to $\lambda_a$ and $\lambda_z$ and the clipping ranges.

**Questions:**

- What are the exact simulator characteristics used for Online World Modeling. Is it deterministic or stochastic. How much wall clock time per rollout. How does performance change as simulator latency increases.
- How many expert trajectories and what diversity were used to train the base teacher forcing model. Please provide dataset sizes per task and the number of finetuning steps for Online and Adversarial methods.
- For Adversarial World Modeling, why choose one step FGSM style perturbations instead of a small number of PGD steps. Did you test two to three steps and if so did robustness and planning performance change.
- How sensitive are results to the perturbation scales. Please include ablations over $\lambda_a$ and $\lambda_z$ and the adaptive radius choice versus fixed radii. Also report the per dimension magnitudes of the perturbations.

---

> ### Author Response · Authors · 2025-11-27
> **Response to Reviewer rdsc - Part 1**
>
> Thank you for your time and feedback! We appreciate that you recognize the simplicity of our approaches along with their evidence for alleviating issues with Gradient-Based Planning and empirical performance gains. We note that in addition to this response post, we’ve also made a separate general post, with several clarifications and new results inspired by your comments. We detail numerous new experiments, analysis, and ablations addressing your points below.
>
> **W1 ("...only three simulated tasks are considered..."):** Inspired by your comments, we evaluate on two robotic manipulation tasks and see that Adversarial World Modeling significantly increases the performance of gradient-based planning and either matches or comes close to the base CEM performance. The tasks are called Rope and Granular, where a simulated XArm is tasked with pushing a piece of rope and a group of one hundred small cubes, respectively. These tasks are more challenging due to a larger number of action dimensions and the finer manipulation required. Furthermore, due to the large cost of the simulator, we only evaluate Adversarial World Modeling on these tasks. We present planning results in terms of Chamfer Distance from the goal set of keypoints (lower is better) in the table below. We have now included these results in Appendix B.2 of the revised submission.
>
> | Method          | Rope GD | Rope CEM |  Granular GD | Granular CEM  |
> |-----------------|---------|----------|--------------|---------------|
> | DINO-WM         | 1.73    | 0.93     | 0.30         | **0.22**      |
> | Adversarial WM  | **0.93**| **0.82** | **0.24**     | 0.28          |
>
> We also evaluate GBP over longer horizons of 50 steps. For PushT and PointMaze, we see that Adversarial World Modeling outperforms DINO-WM on PushT and PointMaze while Online World Modeling outperforms DINO-WM on PointMaze. We present success rate results for long-horizon planning below. Additional details for this experiment can be found in Appendix B.4.
>
> | Long Horizon          | PushT | PointMaze |
> |-----------------|---------|----------|
> | DINO-WM         | 16    | 70     |
> | Online WM         | 16 | **96**|
> | Adversarial WM  | **26**| 88 |
>
> **W2 ("...comparisons to alternative strong planning baselines are incomplete..."):** We focus on settings where explicit state information is not available, instead using high-dimensional image observations to more closely model real environments. Without this explicit state information, iLQR and DDP are computationally infeasible to evaluate. We concur that MPPI is common in these settings and report results for PushT below and in Table 9b of the updated submission. Our implementation is based on [PyTorch-MPPI](https://github.com/UM-ARM-Lab/pytorch_mppi). Additional details for this experiment can be found in Appendix B.5. We also evaluate the hybrid GD + CEM algorithm GradCEM [1] under MPC. This method refines the candidate sequences used to update the estimated action distribution with gradient descent to provide a more accurate estimate of the true distribution's parameters. We share results for GradCEM on PushT below and in Table 9b of the updated submission, finding that Adversarial World Modeling outperforms DINO-WM. Additionally, GradCEM exhibits slightly lower performance than vanilla CEM. We hypothesize this is due to the memory requirements of gradient descent necessitating reducing the number of candidate sequences by a factor of 6 compared to vanilla CEM, leading to reduced accuracy in estimating the true action distribution. Additional details of this experiment can be found in Appendix B.5.
>
> |                  | MPPI | GradCEM |
> |------------------|------|---------|
> | DINO-WM          | 2    | 78      |
> | Online WM        | 2    | 74      |
> | Adversarial WM   | 2    | **84**  |
>
> **W3 (AWM Ablations):** We have conducted several ablations based on your recommendation and address this in detail below in Q4.
>
> **Q1 (Simulator Info):** Each of the tasks we evaluate Online World Modeling on include a deterministic simulator that can be used to reproduce the training trajectories given their actions. We find that rolling out 25 steps in the simulator ranges between 0.7 - 4.5 seconds depending on the task. However, this simulator latency does not affect the number of training steps required for Online World Modeling, which is reported in Appendix A.3. Even with a higher latency simulator for the Wall task, Online World Modeling with Adam comes close to DINO-WM with CEM. We have included further details of the simulators in Appendix B.8.
>
> **Q2 (Training Details):** In Tables 3, 4, and 5 in Appendix A.3, we list the number of unqiue rollouts to train the base DINO-WM model and all training details for OWM and AWM finetuning on each task.

---

> ### Author Response · Authors · 2025-11-27
> **Response to Reviewer rdsc - Part 2**
>
> **Q3 (Number of PGD Steps):** While there are many approaches of adversarial training, we found that both 2-step and 3-step PGD perform on par with FGSM (sometimes surpassing, sometimes degrading), despite requiring 1.5-2x the number of backward passes. Our choice of FGSM over iterative PGD is primarily motivated by its computational efficiency. We have dedicated a section in our "Design Decisions" (Appendix D) to a motivation of our choice of FGSM and report closed-loop planning results on PointMaze and Wall. We also include wall clock time for each epoch.
>
> **Q4 (AWM Perturbations):** We performed a large ablation study varying $\lambda_a, \lambda_z \in [0.0, 0.02, 0.05, 0.20, 0.50, 1.0]^2$ and the choice of adaptive or fixed radii $\epsilon$ to study the sensitivity of AWM to these hyperparameters. For studying the fixed radius, we fix it to the first minibatch's standard deviation and do not update it during training. We found that planning performance is largely unaffected by either of these parameters, except when $\lambda_z = 1.0$. Full results of our grid-search are reported in Appendix D.2. Across experiments, the each dimension perturbation (i.e., action and visual) within their respective perturbation radii $[-\epsilon, \epsilon]$ due to clipping and are roughly zero-mean and very high standard-deviation (close to the maximum possible), suggesting that optimization consistently pushes perturbations toward their limits. As a reference point, in the Wall environment we have $\epsilon_\text{visual} \in [-0.52, 0.52]$ with mean -0.002 and standard deviation 0.45 and $\epsilon_\text{action} \in [-0.15, 0.15]$ with mean 0.004 and standard deviation 0.128.
>
> -------
>
> We believe your feedback has significantly improved our updated draft. We have conducted extensive experiments and made numerous changes to our draft based on your feedback, and we would appreciate it if you would consider raising your score in light of our response. Please let us know if you have additional questions we can address.
>
> [1] Bharadhwaj, H. et al. "Model-Predictive Control via Cross-Entropy and Gradient-Based Optimization." Conference on Learning for Dynamics and Control 2020.
>
> [2] Micheli, V. et al. (2023). "Transformers are Sample-Efficient World Models." International Conference on Learning Representations 2023.

---

### Official Review · Reviewer_PBP2 · 2025-11-05

**Soundness:** 2
**Presentation:** 2
**Contribution:** 2
**Rating:** 4
**Confidence:** 3

**Summary:**

The paper aims to improve gradient-based planning methods. Gradient-based planning methods use a differentiable world model. Given a start observation, goal observation, and planning horizon, they sample an initial set of actions to take and perform stochastic gradient descent to try to find actions that result in reaching the goal from the start observation through the differentiable world model.

The paper proposes two methods to improve gradient based planning: online world modeling and adversarial world modeling. Online world modeling looks at the difference between what the world model predicted and the actual result and finetunes the model based on this. Adversarial world modeling looks for small perturbations in expert trajectories to try to fool the world model into making mistakes and then correcting these mistakes.

The results show that gradient based planning improvements perform better than not using them and in one instance, they are competitive with the CEM method.

**Strengths:**

The paper has a straightforward motivation and gives a clear definition of gradient based planning. The two proposed methods are well justified. The proposed methods show improvements over doing gradient-based planning without them.

**Weaknesses:**

This paper does not define key terms and algorithms and has ambiguity in tables and figures. As a result, someone from a planning background, that does not specialize in planning for robotics, will struggle to understand key points. Here is a lists:
-	The interaction between CEM and other methods is not clear from Table 1.
-	The CEM algorithm for planning is not defined, but it is a key comparison.
-	The planning domains are not defined. Some images of them may help.

The results are only competitive with a non-gradient based method (CEM) in one case. However, it should be noted that gradient-based planning without the proposed modifications is also competitive for this case. Therefore, its hard to say that the methods the authors introduced are responsible for this similarity in performance.

**Questions:**

Why is CEM significant and how does the planning algorithm work?
Is CEM separate from the other methods in Table 1?

---

> ### Author Response · Authors · 2025-11-27
> **Response to Reviewer PBP2**
>
> Thank you for your time and feedback! We appreciate that you recognize the strong justifications for our methods and their improvements on Gradient-Based Planning performance. We clarify key terms and highlight a significant improvement in the competitiveness of our methods below.
>
> **W1 ("...does not define key terms and algorithms..."):** We now provide a detailed explanation of the CEM method in Appendix A.2. The CEM algorithm aims to estimate the distribution of optimal actions by progressively adjusting its mean and standard deviation through iterative sampling from this distribution. As CEM is a gradient-free method, it is a distinct planning algorithm from the gradient-based methods evaluated in Table 1. We have included additional information for each of the environments in Appendix A.1 and images of the tasks in the revised Figure 5.
>
> **W2 (Competitiveness of results):** We have now further tested out a new weighted goal loss as the GBP objective, and found that it improved or maintained planning performance across all tasks. In short, this weighted goal loss also evaluates the distance of intermediate predicted states to the goal, weighted by a sequence of chosen weights. We include details of this loss function in Appendix A.4. With this new GBP objective, **we find that our AWM method is now competitive with or even outperforms CEM on the PointMaze and Wall environments as well as PushT.** We have updated Table 1 in our submission with the latest evaluations and hope this reflects our method's broad improvement.
>
> Additionally, we evaluate on two robotic manipulation tasks and see that Adversarial World Modeling significantly increases the performance of gradient-based planning and either matches or comes close to the base CEM performance. The tasks are called Rope and Granular, where a simulated XArm is tasked with pushing a piece of rope and a group of one hundred small cubes, respectively. These tasks are more challenging due to a larger number of action dimensions and the finer manipulation required. Furthermore, due to the large cost of the simulator, we only evaluate Adversarial World Modeling on these tasks. We present planning results in terms of Chamfer Distance from the goal set of keypoints (lower is better) in the table below. We have now included these results in Appendix B.2 of the revised submission.
>
> | Method          | Rope GD | Rope CEM |  Granular GD | Granular CEM  |
> |-----------------|---------|----------|--------------|---------------|
> | DINO-WM         | 1.73    | 0.93     | 0.30         | **0.22**      |
> | Adversarial WM  | **0.93**| **0.82** | **0.24**     | 0.28          |
>
> We also evaluate GBP over longer horizons of 50 steps. For PushT and PointMaze, we see that Adversarial World Modeling outperforms DINO-WM on PushT and PointMaze while Online World Modeling outperforms DINO-WM on PointMaze. We present success rate results for long-horizon planning below. Additional details for this experiment can be found in Appendix B.4.
>
> | Long Horizon          | PushT | PointMaze |
> |-----------------|---------|----------|
> | DINO-WM         | 16    | 70     |
> | Online WM         | 16 | **96**|
> | Adversarial WM  | **26**| 88 |
>
>
> **Q1 (Significance of CEM):** CEM is important as a predominant search-based algorithm used in planning with world models. Notably, it is a gradient-free method, so it selects from many sampled action sequences to improve the quality of actions, which is prohibitively expensive. Please see W1 for more details regarding the CEM algorithm.
>
> -----------------
>
> We believe your feedback has significantly improved our updated draft. We have conducted extensive experiments and made numerous changes to our draft based on your feedback, and we would appreciate it if you would consider raising your score in light of our response. Please let us know if you have additional questions we can address.

---

### Author Response · Authors · 2025-11-27
**General Response to Reviewers and AC - Part 1**

We thank the reviewers for their valuable feedback. Here we give a general response addressed to all reviewers and ACs, and we address specific reviewer comments in separate posts.

## Novelty and Contribution
To the best of our knowledge, we are the first to propose procedures specifically aimed at improving a world model’s ability to support Gradient-Based Planning (GBP). Despite its computational efficiency, GBP has historically exhibited poor empirical performance, leading prior work to favor sampling-based planners such as CEM. In contrast, our proposed methods enable GBP to consistently match or surpass CEM while achieving over 10× lower wall-clock time (see revised Table 1 and Figure 4). We attribute this substantial improvement to identifying and addressing two core hypotheses underlying GBP’s poor performance. Motivated by reviewer feedback, we have significantly clarified these hypotheses in the revised Introduction, and we now provide stronger empirical support for both. First, we show that Online World Modeling and Adversarial World Modeling reduce the train–test gap in world modeling error, directly improving the accuracy of multi-step rollouts used for planning (revised Section 4.2, Figure 2). Second, we present new visualizations of the action-space loss surface optimized by GBP, demonstrating that Adversarial World Modeling produces a markedly smoother landscape, making gradient-based optimization far more tractable (revised Figure 2). In formulating these methods and studying their effect on GBP performance, we contribute to the understanding of GBP as a whole, a procedure that is often overlooked in the field despite its potential for greatly improving the computational efficiency of planning with world models.

We want to emphasize that the methods we implement were originally developed for other areas, like image classification. We novelly apply techniques that were not intended to be used on world models, namely imitation learning and adversarial training, for the unique approach of finetuning world models to improve GBP performance. We believe that simplicity is a strength, and engineering complex new algorithms is unnecessary when re-purposing simple existing algorithms works well.

## Experimental Breadth
To demonstrate the scalability and applicability of our methods to more challenging and diverse settings, we now evaluate (1) on robotic manipulation tasks, (2) on longer-horizon settings, (3) with alternative planning algorithms, and (4) with a different world model architecture.

**Robotic Manipulation Tasks.** The two robotic manipulation tasks we evaluate on are called Rope and Granular, where a simulated XArm is tasked with pushing a piece of rope and a group of one hundred small cubes, respectively. These tasks are more challenging due to a larger number of action dimensions and the finer manipulation required. Furthermore, due to the large cost of the simulator, we only evaluate Adversarial World Modeling on these tasks. We present planning results in terms of Chamfer Distance from the goal set of keypoints (lower is better) in the table below.

| Method          | Rope GD | Rope CEM |  Granular GD | Granular CEM  |
|-----------------|---------|----------|--------------|---------------|
| DINO-WM         | 1.73    | 0.93     | 0.30         | **0.22**      |
| Adversarial WM  | **0.93**| **0.82** | **0.24**     | 0.28          |

As in PushT, PointMaze, and Wall, we see that Adversarial World Modeling significantly increases the performance of gradient-based planning and either matches or comes close to the base CEM performance. We have now included these results in Appendix B.2 of the revised submission.

**Long Horizon Tasks.** We also evaluate GBP over longer horizons of 50 steps. For PushT and PointMaze, we see that Adversarial World Modeling outperforms DINO-WM on PushT and PointMaze while Online World Modeling outperforms DINO-WM on PointMaze. We present success rate results for long-horizon planning below and in Table 9a of the updated submission.

| Long Horizon          | PushT | PointMaze |
|-----------------|---------|----------|
| DINO-WM         | 16    | 70     |
| Online WM         | 16 | **96**|
| Adversarial WM  | **26**| 88 |

Additional details for this experiment can be found in Appendix B.4.

---

> ### Author Response · Authors · 2025-11-27
> **General Response to Reviewers and AC - Part 2**
>
> **Alternate Planning Methods.** As requested by a few reviewers, we evaluate our methods on different planning algorithms besides GD and CEM. Specifically, we evaluate the Model Predictive Path Integral (MPPI) planning method using an implementation based on [PyTorch-MPPI](https://github.com/UM-ARM-Lab/pytorch_mppi). MPPI is an online, receding-horizon controller that samples and evaluates perturbed action sequences, executes the first action of the lowest-cost trajectory, and then replans from the updated state at each timestep. We share results for PushT below and in Table 9b of the updated submission. Additional details for this experiment can be found in Appendix B.5. We also evaluate the hybrid GD + CEM algorithm GradCEM [1] under MPC. This method refines the candidate sequences used to update the estimated action distribution with gradient descent to provide a more accurate estimate of the true distribution's parameters. We share results for GradCEM on PushT below and in Table 9b of the updated submission, finding that Adversarial World Modeling outperforms DINO-WM. Additionally, GradCEM exhibits slightly lower performance than vanilla CEM. We hypothesize this is due to the memory requirements of gradient descent necessitating reducing the number of candidate sequences by a factor of 6 compared to vanilla CEM, leading to reduced accuracy in estimating the true action distribution. Additional details of this experiment can be found in Appendix B.5.
>
> |                  | MPPI | GradCEM |
> |------------------|------|---------|
> | DINO-WM          | 2    | 78      |
> | Online WM        | 2    | 74      |
> | Adversarial WM   | 2    | **84**  |
>
> **Alternate Architecture.** We also ablate the use of DINO-WM as the base world model architecture by using the IRIS [2] architecture instead. IRIS uses a VQ-VAE encoder and a standard decoder-only transformer for the predictor. We evaluate our methods using the IRIS architecture on the Wall task and find that Adversarial World Modeling makes improvements over base GBP and CEM. Success rate results are below and we have included them in Appendix B.3.
>
> | Method         | GD | CEM |
> |----------------|----|-----|
> | IRIS       | 0  | 4   |
> | IRIS + Online WM      | 0  | 0   |
> | IRIS + Adversarial WM | **8** | **6** |
>
> ## Ablations on Adversarial World Modeling
> On the recommendation of several reviewers, we carefully study various design decisions of our Adversarial World Modeling algorithm including (1) the use of FGSM over 2/3-Step PGD, (2) grid-search over the perturbation scaling factors $\lambda_a, \lambda_z$ and the use of an adaptive or fixed perturbation radii ($\epsilon_a$, $\epsilon_z$), and (3) the role of gradient steps as opposed to clipping. We have dedicated a section in the Appendix titled "Design Decisions" and summarize the findings below.
>
> 1. We found that both 2-step and 3-step PGD perform on par with FGSM (sometimes surpassing, sometimes degrading), despite requiring 1.5-2x the number of backward passes.
>
> 2. We perform grid search with $\lambda_a, \lambda_z \in [0.0, 0.02, 0.05, 0.20, 0.50, 1.0]^2$ and also ablate fixing our radii $\epsilon_a, \epsilon_z$ to their corresponding feature's standard deviation in the \textit{initial minibatch}. This sums to 70 trained adversarial world models. We find that AWM is robust and performant for most scaling factors (with the exception of $\lambda_z = 1.0$) and either the fixed or original adaptive radii selection. Full results of our grid-search are reported in Appendix D.2.
>
> 3. We find that clipping alone with no gradient-step does not produce the same effect as FGSM. In the Wall environment, this produces a finetuned world model with the same performance as the pretrained world model.
>
> [1] Bharadhwaj, H. et al. "Model-Predictive Control via Cross-Entropy and Gradient-Based Optimization." Conference on Learning for Dynamics and Control 2020.
>
> [2] Micheli, V. et al. (2023). "Transformers are Sample-Efficient World Models." International Conference on Learning Representations 2023.

---

### Meta-Review · Area_Chair_zP16 · 2026-01-07

**Summary:**

The reviewers raised several consistent concerns that resulted the rejection recommendation. The primary issue is the limited novelty and strength of the contribution, as the proposed Online World Modeling and Adversarial World Modeling techniques are largely viewed as direct adaptations of well-established methods applied to world model finetuning, without introducing new algorithmic or architectural insights. Additionally, reviewers expressed concerns about the scope and generality of the empirical evaluation. While the authors added experiments in response to feedback, the results remain concentrated on a narrow set of simulated tasks and largely rely on a single world model framework, making it difficult to assess broader applicability. Although the rebuttal addressed many detailed questions and improved presentation, these responses did not substantially change the core concerns regarding novelty, experimental breadth, and contribution level.

**Reviewer Concerns:**

Reviewer Concerns partially addressed by the rebuttal:
1. Experimental completeness and missing details: The authors provided additional experimental details, clarifications of algorithms, and supplementary results (e.g., added tasks, longer-horizon evaluations, and ablations for adversarial training). These additions improved transparency and helped resolve some ambiguities raised by reviewers regarding implementation and evaluation setup.
2. Baseline coverage and additional planners/architectures: In response to reviewer requests, the authors included limited comparisons with additional planners and an alternative world model architecture (IRIS). This partially addresses concerns about narrow evaluation, although the scope remains limited.
3. Design choices in adversarial training: The rebuttal included ablations on perturbation steps, scaling factors, and clipping behavior, clarifying that some heuristic choices were empirically explored rather than arbitrary. This helps justify the implementation but does not fundamentally strengthen the conceptual contribution.

Reviewer Concerns that remains:
1. Limited novelty and incremental contribution (Reviewers P9Wi, yvUu, rdsc): The core concern that the methods are straightforward adaptations of existing techniques (DAgger-style data aggregation and adversarial training) remains. The rebuttal reframes the contribution but does not introduce new algorithmic insights, theory, or principled justification beyond empirical tuning.
2. Experimental scope and generality: Despite added experiments, evaluations remain confined to a small set of simulated tasks and are largely tied to a specific world model setup. Concerns about generalization to diverse environments, harder long-horizon tasks, or real-world settings are not convincingly resolved.

**Reviewer Scores:**

Reviewer PBP2,
Original score: 4.
The rebuttal clarifies several definitions and adds experiments, but concerns about competitiveness with CEM and attribution of gains are not fully resolved.
Score after discussion: 4.

Reviewer rdsc,
Original score: 6.
While the rebuttal addresses many specific questions with additional experiments and ablations, core concerns about limited scope, heuristic design choices, and overall contribution remain.
Score after discussion: 6 or 4 (likely unchanged or slightly lower).

Reviewer P9Wi,
Original score: 2.
Strong concerns regarding incremental novelty.
Score after discussion: 2.

Reviewer yvUu,
Original score: 2.
Concerns about lack of novelty.
Score after discussion: 2.

---

### Decision · Program_Chairs · 2026-01-26

Reject